# STORY-ITER: A TRAINING-FREE ITERATIVE PARADIGM FOR LONG STORY VISUALIZATION

**Jiawei Mao**[1*]  **Xiaoke Huang**[1*]  **Yunfei Xie**[2]  **Yuanqi Chang**[1]  **Mude Hui**[1]
**Bingjie Xu**[3]  **Zeyu Zheng**[4]  **Zirui Wang**[5]  **Cihang Xie**[1]  **Yuyin Zhou**[1]

[1]UC Santa Cruz    [2]Rice University    [3]Singapore Institute of Technology
[4]UC Berkeley    [5]Apple

🌐 **Project Page**: `https://github.com/UCSC-VLAA/story-iter`

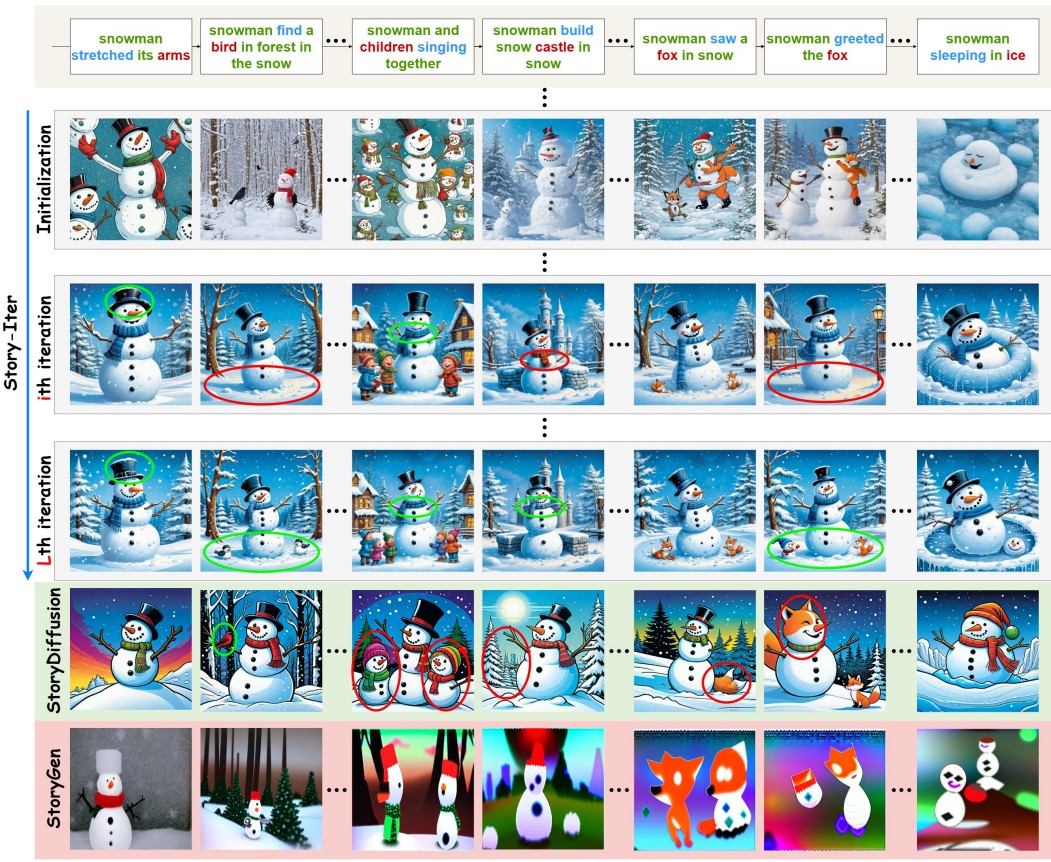

Figure 1: A long story of "*snowman*" visualized by our Story-Iter from different iterations, compared with those visualized by previous StoryDiffusion Zhou et al. (2024) and StoryGen Liu et al. (2024). Notable differences are highlighted in green and red. Zoom in for a better view.

## ABSTRACT

This paper introduces **Story-Iter**, a new training-free iterative paradigm to enhance long-story generation. Unlike existing methods that rely on fixed reference images to construct a complete story, our approach features a novel external **iterative paradigm**, extending beyond the internal iterative denoising steps of diffusion models, to continuously refine each generated image by incorporating all reference images from the previous round. To achieve this, we propose a plug-and-play, training-free **g**lobal **r**eference **c**ross-**a**ttention (**GRCA**) module, modeling all reference frames with global embeddings, ensuring semantic consistency in long sequences. By progressively incorporating holistic visual context and text constraints, our iterative paradigm enables precise generation with fine-grained in-

---

[*]Equal contribution.

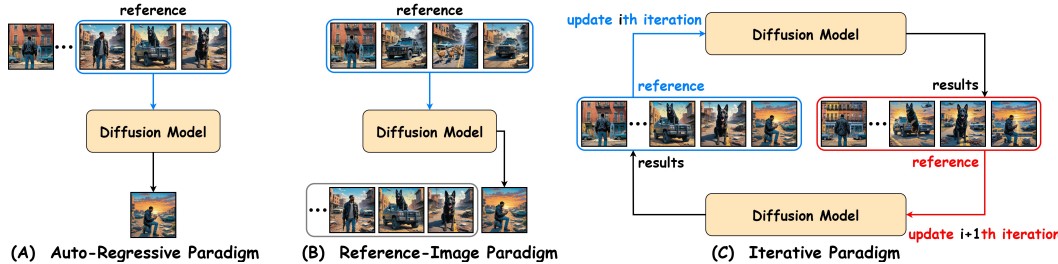

Figure 2: Comparison of paradigms for long story visualization: (A) Auto-Regressive (AR): generates frames sequentially referencing previous finite frames (*e.g.* the previous three frames); (B) Reference-Image (RI): employs fixed reference images (*e.g.* the beginning four frames) as reference images; (C) Iterative Paradigm: leverages all frames from the previous iteration as reference images.

teractions, optimizing the story visualization step-by-step. Extensive experiments in the official story visualization dataset and our long story benchmark demonstrate that Story-Iter's state-of-the-art performance in long-story visualization (up to *100 frames*) excels in both semantic consistency and fine-grained interactions.

# 1 INTRODUCTION

Story visualization aims to generate a sequence of coherent images from text prompts, reflecting the narrative's progression and enabling users, even without an artistic background, to visually present their stories (Li et al., 2019; Maharana & Bansal, 2021; Chen et al., 2022). Recent advancements in text-to-image models, particularly diffusion models, have significantly improved the quality of generated visuals, producing high-quality, creative, and aesthetically pleasing images (Saharia et al., 2022; Rombach et al., 2022; Kang et al., 2023). These models greatly outperform earlier approaches, such as generative adversarial networks (Brock, 2018) in terms of image quality.

However, story visualization remains challenging, particularly in maintaining semantic consistency and synthesizing interactions as the story length increases. Two main paradigms have emerged in this domain. The **Auto-Regressive Paradigm** (Fig. 2 (A)), which generates frames sequentially referencing previous finite frames (Pan et al., 2024; Liu et al., 2024), often struggles with semantic consistency due to error accumulation and the inability to reference future frames. Although techniques like Consistent Self-Attention (CSA) (Zhou et al., 2024) can help mitigate these inconsistencies, their reliance on intermediate denoising features results in high memory consumption, limiting scalability for longer stories. To address these challenges, Zhou et al. (2024) further proposes the **Reference-Image Paradigm** (Fig. 2 (B)), which employs fixed reference images to guide the visualization. However, while using only the initial frames as reference images alleviates scalability issues, it fails to provide the global semantic coherence necessary for long-story visualization, resulting in error propagation from the reference images to subsequent frames. As such, both paradigms experience quality degradation when visualizing long stories. Additionally, they inherit the limitations from Stable Diffusion Model (SDM) (Rombach et al., 2022), particularly in generating fine-grained interactions in the story (Fig. 1).

To address the limitations in existing story generation methods, we present Story-Iter, a new **Iterative Paradigm**. Unlike existing methods that use multiple fixed reference frames but do not refine them over time or incorporate a holistic visual context (Fig. 2 A&B, *e.g.* StoryGen (Liu et al., 2024) and StoryDiffusion (Zhou et al., 2024)), our approach continuously refines each generated story frame, utilizing the full-length reference images generated from the previous iteration. Here, iteration refers to the external round beyond the internal denoising steps of diffusion models.

The proposed iterative paradigm offers two key advantages. 1) It progressively approximates the distribution of reference image global embeddings across iterations, ultimately ensuring visual coherence (see Fig. 4). 2) By iteratively engaging the full-length story frames and text prompts, it optimizes fine-grained control based on both global and local semantic contexts. Shown in Fig. 1, Story-Iter enhances both visual consistency and fine control across iterations, resulting in more coherent and higher-quality visualizations. For example, the image depicting complex character

interactions, such as "snowman saw a fox" demonstrates substantial improvement over iterations compared to previous methods (Liu et al., 2024; Zhou et al., 2024).

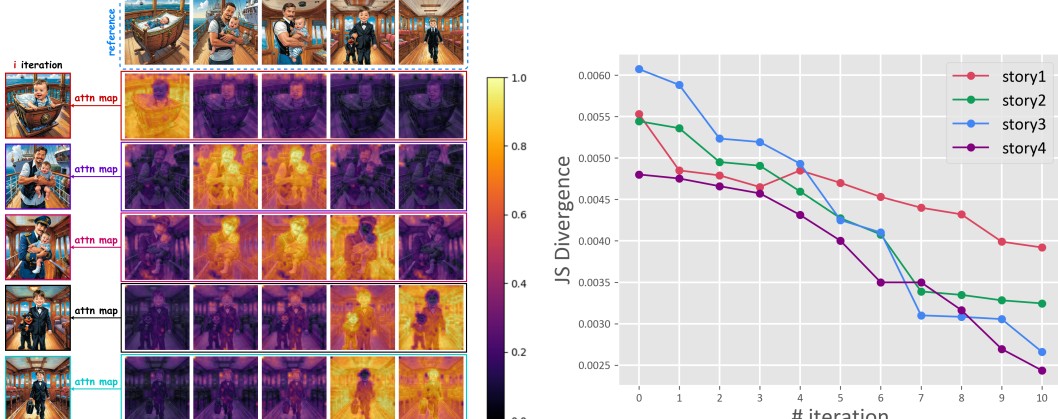

Figure 3: Attention maps for each generated image in our Global Reference Cross-Attention. Zoom in for a better view.

Figure 4: Distribution among the global embeddings of full-length reference images as iterations progress.

Specifically, during initialization, only text prompts of the story are utilized to generate the full story frames without any reference. In the subsequent iterations, the global embeddings of the full-length story frames from the previous iteration as the reference, along with the text embeddings, collaboratively guide story generation. To implement the iterative paradigm, we introduce a plug-and-play *Global Reference Cross-Attention (GRCA)* for long story visualization. Some subject-consistent generation methods, such as the decoupled cross-attention in IP-Adapter (Ye et al., 2023), also contribute to story visualization. However, they are designed to maintain semantic consistency in individual images, whereas story visualization requires connections across all frames, making them not directly applicable. Different from the IP-Adapter, the global embeddings of the full-length reference images act as keys and values in GRCA. Unlike the intermediate denoising features used in CSA in StoryDiffusion (Zhou et al., 2024), the global embeddings in GRCA allow more image frames to be referenced to maintain global semantic consistency. The processes of computing the token similarity matrix and merging tokens in GRCA adaptively aggregate visual features from reference images similar to the currently generated image (Fig. 3). This ultimately ensures the semantic consistency in the iterative paradigm and reduces the influence from noisy references. Additionally, to strike a balance between visual consistency and text controllability, we introduce a linear weighting strategy to fuse both modalities.

Extensive experiments demonstrate that Story-Iter consistently outperforms existing methods for visualizing both regular-length and long stories (up to *100 frames*). Specifically, in the context of regular-length story visualization using the StorySalon benchmark (Liu et al., 2024), Story-Iter exceeds the baseline model, StoryGen (Liu et al., 2024), achieving a 9.4% improvement in average Character-Character Similarity (aCCS) (Cheng et al., 2024b) and a 21.71 reduction in average Fréchet Inception Distance (aFID) (Cheng et al., 2024b). For long story visualization, Story-Iter also demonstrates solid advancements, achieving gains of 3.4% in aCCS and 8.14 in aFID compared to StoryDiffusion (Zhou et al., 2024), demonstrating the superior generative quality of Story-Iter, particularly in semantic consistency and fine-grained interactions.

Our contributions are summarized as follows:

- **A new long story benchmark** for evaluating long story visualization for up to **100 frames**.

- **A new iterative paradigm** that enhances story consistency by continuously updating reference images in each *external iteration*, beyond the diffusion model's internal denoising steps.

- **A new global attention mechanism GRCA** that enables modeling all frames as reference images, thus ensuring semantic consistency in long sequences.

- **State-of-the-art** story visualization performance on various benchmarks, especially in long-story scenarios.

## 2 RELATED WORK

**Story Visualization** Story visualization (Chen et al., 2022; Li, 2022; Tao et al., 2025; Papadimitriou et al., 2024; Bugliarello et al., 2024; Ahn et al., 2023; Rahman et al., 2023; Tsakas et al., 2023; Wu et al., 2024) has evolved from GAN-based approaches like StoryGAN (Li et al., 2019) to more advanced techniques. Recent developments leverage diffusion models (Shen et al., 2024; Tao et al., 2024) and combine them with the auto-regressive paradigm, as seen in AR-LDM (Pan et al., 2024) and StoryGen (Liu et al., 2024). StoryDiffusion (Zhou et al., 2024), on the other hand, uses a reference-image paradigm with fixed reference images. DreamStory (He et al., 2024), StoryGPT (Shen & Elhoseiny, 2023), and MovieDreamer (Zhao et al., 2024) introduce LLM-generated guidance for story visualization. There are also attempts on interactive story visualization (Cheng et al., 2024a;b; Gong et al., 2023; Yang et al., 2024). However, challenges remain in maintaining semantic consistency for the whole story and avoiding error accumulation, especially for longer narratives (Wang et al., 2023; Tewel et al., 2024; Zhou et al., 2024; Liu et al., 2024). Unlike StoryDiffusion (Zhou et al., 2024), which uses fixed reference frames but does not refine them or incorporate holistic visual context, our paradigm continuously refines each generated frame, utilizing the full-length story frames generated from the previous external iteration.

## 3 METHOD

### 3.1 INITIALIZATION

To build the initialization for iteration, we only employ text prompt $T_k$ for the $k_{th}$ image in the story to guide the pretrained Stable Diffusion Model (Rombach et al., 2022) $\mathrm{SDM}(z, T_k)$ in generating the initial images, where $z$ is the random noise. All generated images from the initial step will be stored as reference images for the first iteration. We denote $i = 0$ as the initialization of Story-Iter. Thus, the initially generated story frames can be represented as:

$$
\begin{aligned}
x_k^{i=0} &= \mathrm{SDM}(z, T_k),\ k \in [1,\ B], \\
x_{1 \cdots B}^{i=0} &= [x_1^{i=0},\ x_2^{i=0}, \cdots, x_k^{i=0}, \cdots, x_{B-1}^{i=0},\ x_B^{i=0}],
\end{aligned}
\tag{1}
$$

where $B$ denotes the story length. Compared to subject-consistent image generation methods (Ye et al., 2023) that introduce reference image guidance, Story-Iter is initialized only by text prompts to outline the story faithfully.

### 3.2 ITERATIVE PARADIGM

In this subsection, we describe how each image is updated within an external iteration beyond the internal denoising steps of diffusion models (see Fig. 5). Formally, for the $i_{th}$ iteration, we use the full-length $B$ story frames from the $(i-1)_{th}$ iteration $x_{1 \cdots B}^{i-1}$ ($i \in [1,\ L]$, $L$ refers to total iteration number) as the reference images to refine story visualization. With $\mathrm{SDM}_{\mathrm{GRCA}}(z, T_k, x_{1 \cdots B}^{i-1})$ representing the whole denoising process with our Global Reference Cross-Attention in pretrained Stable Diffusion Model (Sec. 3.3), the $k_{th}$ story frame generated from the $i_{th}$ iteration can be expressed as:

$$
\begin{aligned}
x_k^i &= \mathrm{SDM}_{\mathrm{GRCA}}(z, T_k, x_{1 \cdots B}^{i-1}),\ k \in [1,\ B], \\
x_{1 \cdots B}^i &= [x_1^i,\ x_2^i, \cdots, x_k^i, \cdots, x_{B-1}^i,\ x_B^i],
\end{aligned}
\tag{2}
$$

As the iterations progress, the distribution of reference image global embeddings gradually converges, thereby assisting Story-Iter in continuously enhancing the semantic consistency of the global view. As shown in Fig. 4, Story-Iter does not enforce consistency in a single step. Instead, through successive external iterations, it **progressively refines GRCA** by aligning the current frame's features with semantically relevant content from reference frames. Fine-grained interactions (*limitation for SDM*) are also optimized as Story-Iter repeatedly engages text constraints $T_k$ in the iterative paradigm.

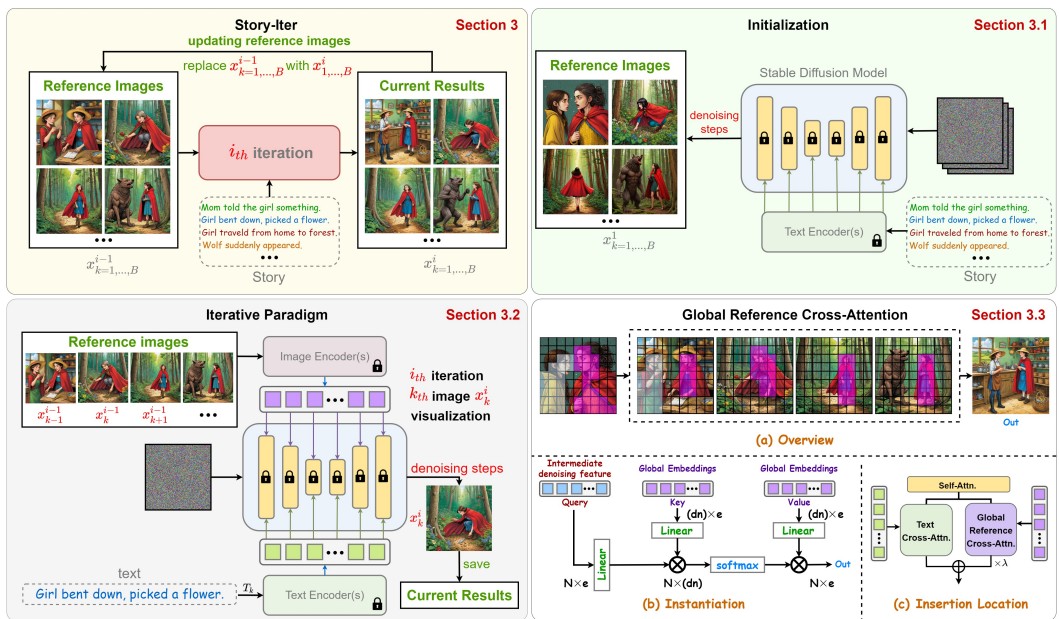

Figure 5: Overview of Story-Iter. The full-length story frames are generated with only text prompts during initialization (Sec. 3.1). In the subsequent external iterations, the story frames are generated referencing the full-length story frames from the previous iteration (Sec. 3.2). The Global Reference Cross-Attention (GRCA) adaptively aggregates features from full-length reference images to ensure semantic consistency (Sec. 3.3).

## 3.3 GLOBAL REFERENCE CROSS-ATTENTION

We propose a plug-and-play augmentation module to equip Stable Diffusion models, called Global Reference Cross-Attention (GRCA). We utilize pre-trained CLIP (Radford et al., 2021) as the image encoder and project the global embedding (termed $c$) for each reference image into a few tokens. This global embedding enables guidance of structural and visual similarity to reference images during the denoising process. Furthermore, with fewer tokens, GRCA can incorporate more reference images.

Given full-length $B$ images generated from the $(i-1)_{th}$ iteration $x_{1\cdots B}^{i-1} \in \mathbb{R}^{B \times h \times w \times 3}$ ($h$: height, $w$: width), we define $\text{Attention}(Q, K, V)$ to indicate the attention calculation, where $Q$, $K$, and $V$ represent the query, key, and value respectively. GRCA of the $k_{th}$ image in the $i_{th}$ iteration:

$$
\begin{aligned}
c_{1\cdots B}^i &= \text{CLIP}(x_{1\cdots B}^{i-1}), \quad c_{1\cdots B}^i \in \mathbb{R}^{B \times d}, \\
\hat{c}_{1\cdots B}^i &= c_{1\cdots B}^i W_c, \quad W_c \in \mathbb{R}^{d \times (n \times e)}, \\
\tilde{c}_{1\cdots B}^i &= \text{flatten}(\hat{c}_{1\cdots B}^i), \quad \tilde{c}_{1\cdots B}^i \in \mathbb{R}^{1 \times (B \times n) \times e}, \\
Q_k^i &= I_k W_q, \ K_k^i = \tilde{c}_{1\cdots B}^i W_k, \ V_k^i = \tilde{c}_{1\cdots B}^i W_v, \\
\text{GRCA}(I_k^i, x_{1\cdots B}^{i-1}) &= \text{Attention}(Q_k^i, K_k^i, V_k^i).
\end{aligned}
\tag{3}
$$

The dimension of global embedding $c_k^i$ and the projected dimension are denoted by $d$ and $e$, respectively. $n$ indicates the length of reference tokens, $n = 4$ if not specified. $\text{flatten}(.)$ represents a flatten operation on vectors. $W_c$ is the projection matrix transforming the global embeddings into reference tokens. $W_q$ is the weight matrix of the intermediate denoising feature $I_k^i$ in SDM. $W_k$ and $W_v$ are the weight matrices of the reference tokens. The proposed plug-and-play GRCA directly reuses the cross-attention weights from IP-Adapter (Ye et al., 2023) in subject-consistent single-image generation without training. Finally, we merge the outputs from GRCA with the outputs from cross-attention on texts, to guide the $k_{th}$ story frame generation. With corresponding text prompt $T_k$ and the full-length reference images $x_{1\cdots B}^{i-1}$, the intermediate denoising feature $I_k^i$ is updated as:

$$
I_k^i = \text{Attention}(I_k^i, T_k, T_k) + \lambda_i \text{GRCA}(I_k^i, x_{1\cdots B}^{i-1})
\tag{4}
$$

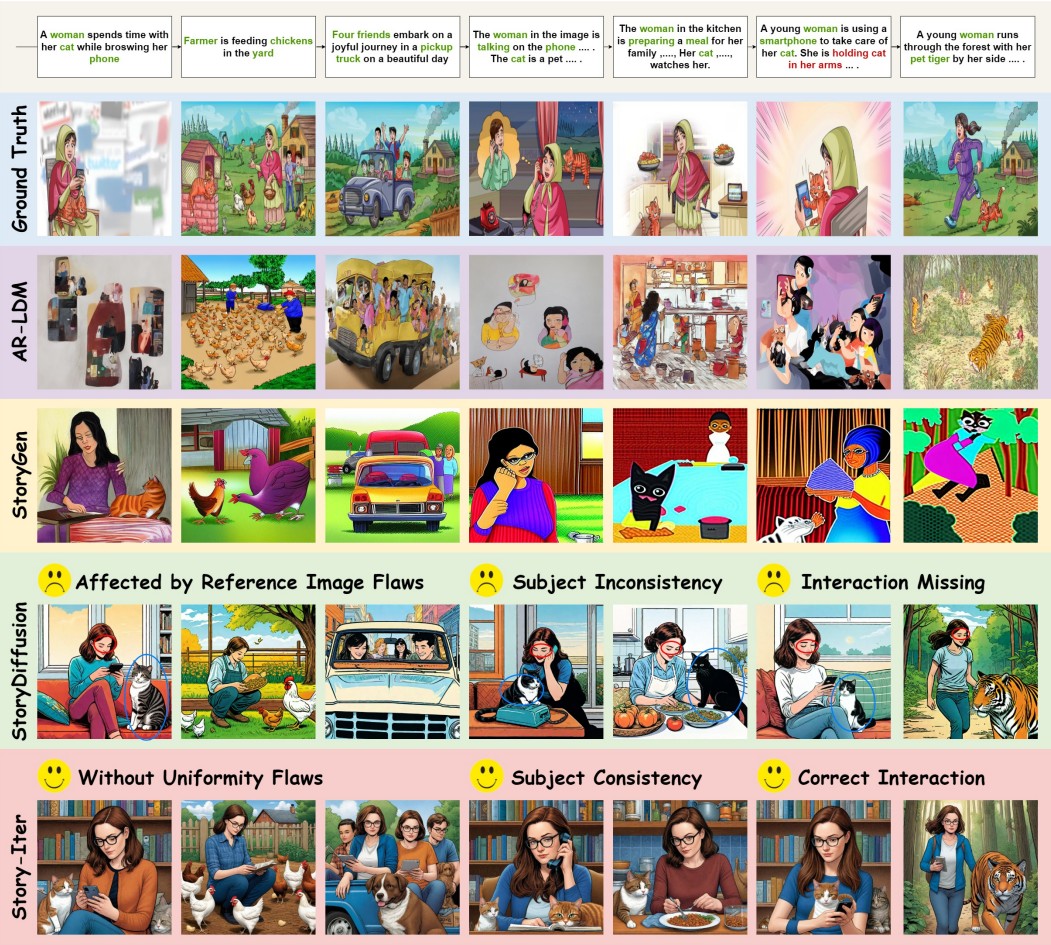

Figure 6: Qualitative comparisons for *regular-length* story visualization. Zoom in for a better view.

where the weight factor $\lambda_i$ of the $i_{th}$ iteration is adjusted as $\lambda_i = \lambda_1 + q \times (i - 1)$ with $q$ fixed. $\lambda_i$ increases linearly with a lower bound $\lambda_1$ to balance visual consistency and text alignment in the iterative paradigm. Leveraging pretrained weights, GRCA adaptively re-weights attention at each iteration, thereby mitigating the accumulation of noise from irrelevant historical states and instead prioritizing information most pertinent to the current frame (see Fig. 3). The global embedding undergoes progressive updates, serving as a soft forgetting mechanism that attenuates contextual drift and naturally suppresses obsolete or extraneous information throughout the iterative refinement process. We demonstrate *Pseudo-code* of Story-Iter in Algo. 1.

## 4 EXPERIMENTS

### 4.1 EXPERIMENTAL SETTING

We use the StorySalon dataset (Liu et al., 2024) to benchmark performance for *regular-length* story visualization. Compared to the closed Flintstones (Gupta et al., 2018) and Pororo (Kim et al., 2017) benchmarks, the open-ended StorySalon benchmark provides an evaluation of more diverse scenarios. For *long* story visualization, we curate multiple long stories using GPT-4o (OpenAI, 2024). To evaluate the efficacy of Story-Iter, we report CLIP text-image similarity (CLIP-T) (Radford et al., 2021), average Fréchet Inception Distance (aFID) (Cheng et al., 2024b), and Character-Character Similarity (aCCS) (Cheng et al., 2024b). CLIP-T is to measure image-text alignment, and both aFID and aCCS are used to evaluate semantic consistency among generated images. We use 10 iterations and the fixed random seed for Story-Iter in experiments by default. For story visualization within 100 frames, 10 iterations are sufficient to maintain the consistency of the story sequence. Iteration number ablation in Sec. F. *Human evaluation* and *diversity* can be found in Sec. C and Sec. H.3.

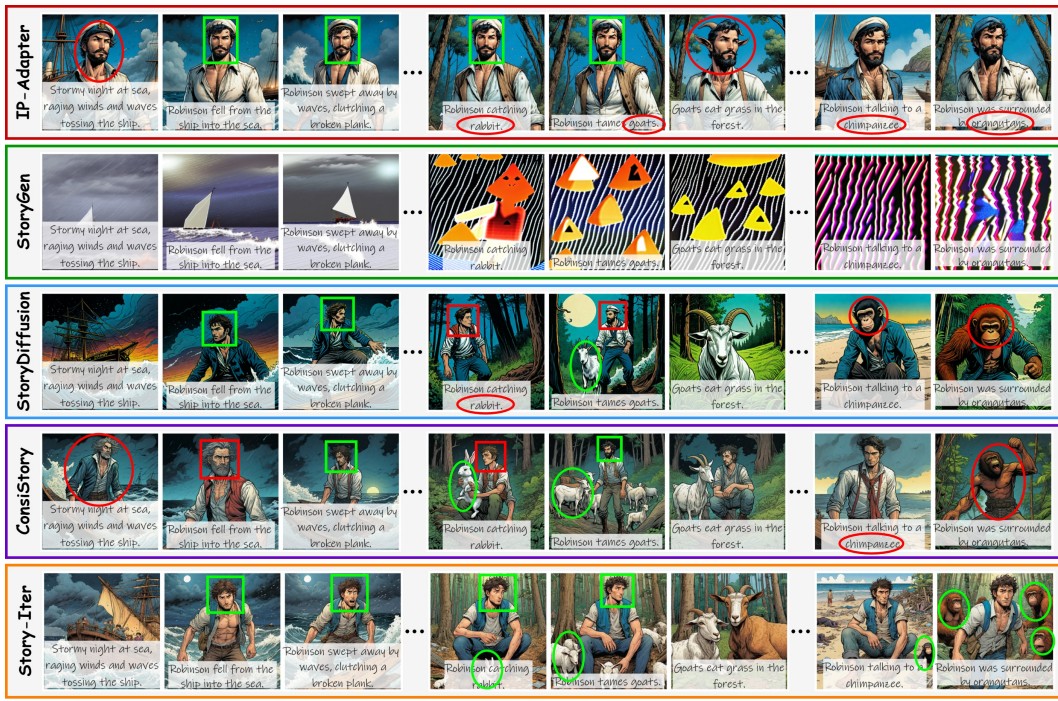

Figure 7: Qualitative comparisons for *long* story visualization. Story-Iter shows advantages in generating semantic consistency (consistency and differences are highlighted in green and red boxes) and character interactions (correct and incorrect interactions are highlighted in green and red circles). For the complete story, refer to the Fig. 24.

Table 1: Quantitative comparison for *regular-length* stories.

| Method | CLIP-T ↑ | aCCS ↑ | aFID ↓ |
|---|---|---|---|
| SDM (Rombach et al., 2022) | **0.323** | 0.662 | 23.10 |
| Prompt-SDM (Rombach et al., 2022) | 0.289 | 0.699 | 18.18 |
| Finetuned-SDM (Rombach et al., 2022) | 0.309 | 0.639 | 23.05 |
| AR-LDM (Pan et al., 2024) | 0.239 | 0.684 | 42.55 |
| StoryGen (Liu et al., 2024) | 0.255 | 0.724 | 36.34 |
| IP-Adapter (Ye et al., 2023) | 0.241 | 0.737 | 25.18 |
| Story-Iter (Ours) | 0.305 | 0.760 | 16.52 |
| IP-AdapterXL (Ye et al., 2023) | 0.244 | 0.758 | 14.70 |
| StoryDiffusion (Zhou et al., 2024) | 0.311 | 0.765 | 14.84 |
| Story-IterXL (Ours) | 0.310 | **0.818** | **14.63** |

Table 2: Quantitative comparison for *long* story visualization.

| Method | CLIP-T ↑ | aCCS ↑ | aFID ↓ |
|---|---|---|---|
| AR-LDM (Pan et al., 2024) | 0.216 | 0.673 | 133.62 |
| StoryGen (Liu et al., 2024) | 0.223 | 0.740 | 126.13 |
| IP-Adapter (Ye et al., 2023) | 0.274 | 0.751 | **93.70** |
| Story-Iter (Ours) | **0.307** | **0.754** | 98.51 |
| FLUX.1.Kontext (Batifol et al., 2025) | 0.290 | 0.783 | **78.03** |
| Nano Banana | 0.310 | 0.791 | 106.14 |
| IP-AdapterXL (Ye et al., 2023) | 0.297 | 0.787 | 88.69 |
| DALL·E 3 (Betker et al., 2023) | 0.301 | 0.674 | 251.71 |
| SEED-LLaMA (Ge et al., 2023) | 0.279 | 0.688 | 88.96 |
| ConsiStory (Tewel et al., 2024) | 0.316 | 0.761 | 108.83 |
| 1prompt1story (Liu et al., 2025) | 0.316 | 0.790 | 99.25 |
| StoryDiffusion (Zhou et al., 2024) | 0.315 | 0.768 | 102.44 |
| Story-IterXL (Ours) | **0.318** | **0.802** | 94.30 |

## 4.2 REGULAR-LENGTH STORY VISUALIZATION

Based on the standard setup on StorySalon dataset (Liu et al., 2024), we compare with the following baselines: StoryDiffusion (Zhou et al., 2024), StoryGen (Liu et al., 2024), AR-LDM (Pan et al., 2024), SDM (Rombach et al., 2022), Prompt-SDM (Rombach et al., 2022), Finetuned-SDM (finetuned on StorySalon) (Rombach et al., 2022), and IP-Adapter (individual image generation) (Ye et al., 2023). Autostudio (Cheng et al., 2024a) requires the additional bounding boxes as prompt, thus it is not included in comparison. For Prompt-SDM, we use prompts of "cartoon-style images". For StoryDiffusion, we follow the official setup by using the initial four generated images as the reference. To adhere to copyright restrictions and ensure fair comparisons, we exclusively utilize text prompts from the open-source subset of the StorySalon test set for evaluation. This subset comprises 6,026 prompts, with an average of 14 frames per story and the longest story up to 44 frames.

**Quantitative Evaluation.** CLIP-T results in Tab. 1 show that Story-Iter and StoryDiffusion (Zhou et al., 2024) visualize content more aligned to the text prompt than previous story visualization models (AR-LDM (Pan et al., 2024) and StoryGen (Liu et al., 2024)). Since the story visualization

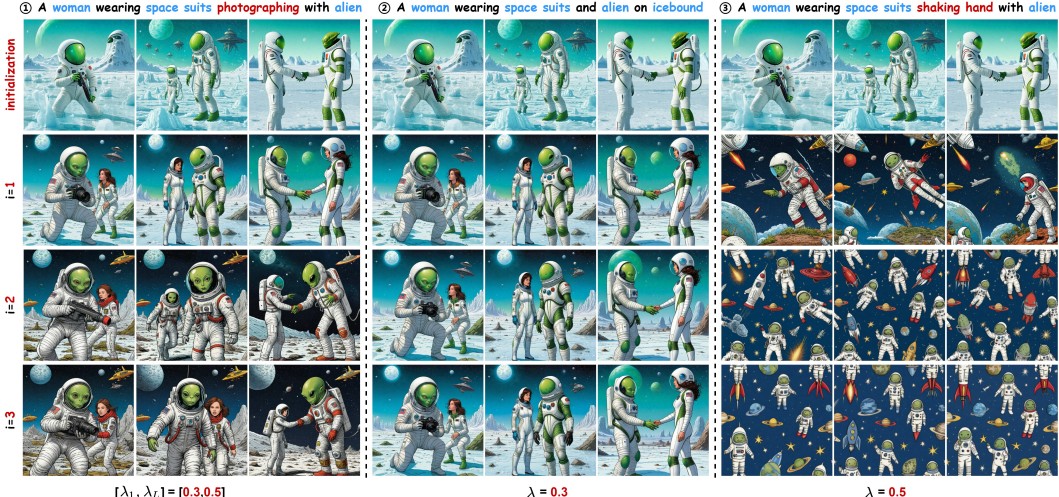

Figure 8: Ablation study of the weighting strategies in the iterative paradigm: linear weighting strategy $\lambda_i = \lambda_1 + q \times (i-1)$ for the $i_{th}$ iteration or fixed weighting strategy (*e.g.* $\lambda = 0.3$, $\lambda = 0.5$). Zoom in for a better view. See Sec. F for results with more iterations.

framework introduces additional image conditioning, it inevitably affects text alignment compared to SDM, which is conditioned only on text. Meanwhile, since neither Story-Iter nor most baselines are trained on the StorySalon dataset, we introduce aFID and aCCS metrics for a fair evaluation of the character consistency among generated story images. aFID and aCCS in Tab. 1 illustrate that Story-Iter achieves higher semantic consistency of the generated images compared to most baselines.

**Qualitative Evaluation.** In Fig. 6, we provide the qualitative comparison results of the open-ended story visualization. Although AR-LDM (Pan et al., 2024) and StoryGen (Liu et al., 2024) generate coherent image sequences based on story prompts, the quality of the generated images degrades when story length increases due to the error accumulation issue of the AR paradigm. Results of StoryDiffusion (Zhou et al., 2024) and Story-Iter show satisfactory story visualization performance. However, StoryDiffusion cannot maintain consistency between certain subjects due to lacking global story comprehension (*e.g.*, "*cat*" in Fig. 6). Additionally, since StoryDiffusion requires the first few generated images as references, the visualization results are affected by the reference image flaws (*e.g.*, "*closed-eye issue*" in Fig. 6). In comparison, Story-Iter outperforms in regular-length story visualization by generating coherent image sequences. These findings highlight the effectiveness of GRCA, as it incorporates all reference images to ensure semantic consistency.

### 4.3 LONG STORY VISUALIZATION

To better evaluate generative quality for long story visualization (*i.e.*, up to 100 frames (Fig. 32 and Fig. 33)), we compare to existing story visualization methods, reference-image based image editing models, and reference-guided image generation methods. We use GPT-4o to generate 20 long story cases of 10 50-sentence and 10 100-sentence narratives.

**Quantitative Evaluation.** The quantitative results in Tab. 2 show that Story-Iter significantly improves the semantic consistency and the generative coherence for fine-grained interactions for long story visualization compared to existing models (Pan et al., 2024; Liu et al., 2024; Betker et al., 2023; Ge et al., 2023; Tewel et al., 2024; Liu et al., 2025). A key difference between our Story-Iter and StoryDiffusion (Zhou et al., 2024) lies in the paradigm: by refining frame generation with global (visual) and local (text) semantic context in the iteration paradigm, Story-Iter achieves higher consistency compared to StoryDiffusion, which relies on the fixed reference images. IP-Adapter (Ye et al., 2023) employs the same single reference image that leads to less aFID. In contrast, referencing full-length frames in our paradigm, our method ensures both visual consistency and text alignment. Compared with Story-Iter, reference-image editing models suffer from overly strong identity constraints and their inability to handle newly introduced characters, which makes them unsuitable for long-story visualization. See Sec. D for more comparisons with reference-image editing models.

Table 3: Comparison of computational efficiency across different Story-Iter variants for *100* frames *1024×* story visualization.

| Method | Diffusion Steps | External Iterations | FLOPs | VRAM | Time | CLIP-T | aCCS | aFID |
|---|---|---|---|---|---|---|---|---|
| StoryDiffusion (Zhou et al., 2024) | 50 | 1 | 22 PFLOPs | 40GB | 31 min | 0.315 | 0.768 | 102.44 |
| Story-IterXL | 50 | 10 | 43 PFLOPs | 19GB | 250 min | 0.318 | 0.802 | 94.30 |
| Story-IterXL-Fast | 4 | 10 | 3 PFLOPs | 19GB | 20 min | 0.309 | 0.788 | 109.13 |

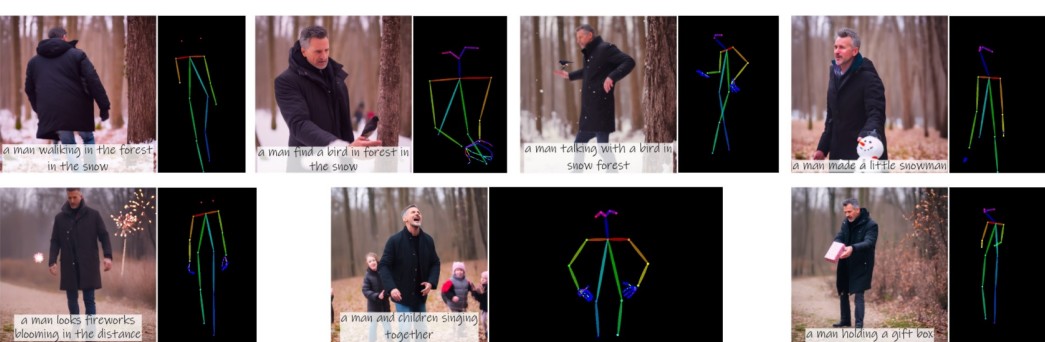

Figure 9: Pose-conditioned generation results of Story-Iter-ControlNet. Given pose maps, the model produces frames that closely match the target poses while maintaining global story coherence.

**Qualitative Evaluation.** Fig. 7 shows the visualization results for long stories, indicating that Story-Iter can generate high-quality, thematically consistent long image sequences based on the text prompts. The results of the IP-Adapter (Ye et al., 2023) exhibit significant missing or incorrect interactions due to limited text controllability. Our method outperforms IP-Adapter in global context visualization, benefiting from the proposed iterative paradigm and referencing full-length story-frames, in contrast to single-image guided generation. StoryGen (Liu et al., 2024) suffers from a more serious error accumulation issue from the AR paradigm during extended storytelling. Without modeling the holistic visual context, StoryDiffusion (Zhou et al., 2024) and ConsiStory (Tewel et al., 2024) fail to generate faithful character interactions and maintain consistency throughout the story. Our Story-Iter refers to the full-length story frames and individual text prompts in external iterations, thus achieving more accurate character interactions and maintaining subject consistency.

### 4.4 VARIANTS

**Story-Iter-Fast.** To improve the computational efficiency of Story-Iter for long-story visualization, we develop a fast variant built upon the SDXL-LCM backbone (Luo et al., 2023). This version reduces the diffusion sampling steps from 50 to only 4, substantially accelerating the iterative generation process while maintaining story-level consistency comparable to the original Story-Iter (see Tab. 3 and Sec. H.2).

**Story-Iter-ControlNet.** We introduce a new variant, Story-Iter-ControlNet, which integrates a pose-conditioned ControlNet (Zhang et al.) into the Story-Iter framework. This variant enables explicit control over character pose and spatial layout while fully preserving the global story consistency and text–image alignment achieved by Story-Iter. As shown in Fig. 9, by using pose maps (e.g., OpenPose skeletons) as control signals, Story-Iter-ControlNet enhances local-frame diversity without compromising global consistency or text–image alignment.

### 4.5 ABLATION STUDY

**Global Reference Cross-Attention.** We ablate the effect of global semantics modeling by GRCA for long story visualization. Specifically, for each image visualization in the sequence, we only use the single reference image at the corresponding index during the iteration as guidance. By establishing a global comprehension of the story for the diffusion model, Story-Iter maintains the semantic consistency in the generated image sequence (Tab. 4 and Fig. 10). For further discussion on GRCA, please refer to the Sec. A.5.

**Iterative Paradigm.** We conduct ablation experiments to evaluate the effect of the proposed iterative paradigm for long story visualization. As shown in Tab. 4 and Fig. 8, the iterative paradigm improves

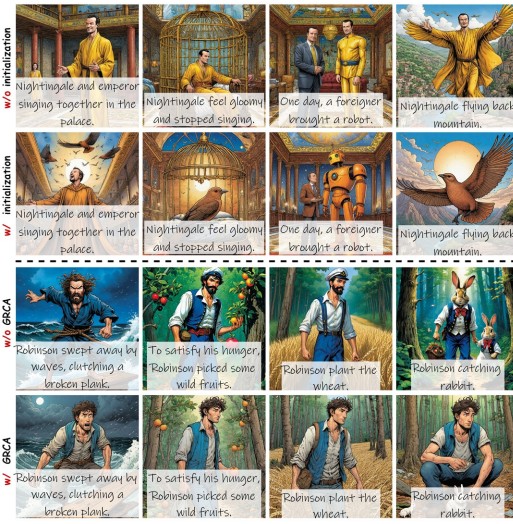

Figure 10: Qualitative ablation studies of initialization and GRCA. Zoom in for a better view.

Table 4: Ablation of the design choices of Story-Iter.

| Setting | CLIP-T ↑ | aCCS ↑ | aFID ↓ |
|---|---|---|---|
| *w/o* Initialization | 0.302 | 0.788 | **90.30** |
| *w/o* GRCA | 0.319 | 0.740 | 97.86 |
| *w/o* Iteration Paradigm | **0.322** | 0.757 | 105.17 |
| **Ours** | 0.318 | **0.802** | 94.30 |

Table 5: Effect of different weighting strategies in the Story-Iter. $[\lambda_1, \lambda_L]$ denotes a linear weighting strategy with a lower bound of $\lambda_1$ and an upper term of $\lambda_L$ in 10 iterations.

| *fixed weighting strategy* | CLIP-T ↑ | aCCS ↑ | aFID ↓ |
|---|---|---|---|
| $\lambda = 0.3$ | 0.320 | 0.760 | 101.55 |
| $\lambda = 0.5$ | 0.261 | 0.753 | 81.72 |

| *linear weighting strategy* | CLIP-T ↑ | aCCS ↑ | aFID ↓ |
|---|---|---|---|
| $[\lambda_1, \lambda_L] = [0.2, 0.5]$ | 0.320 | 0.781 | 101.44 |
| $[\lambda_1, \lambda_L] = [0.3, 0.4]$ | 0.318 | 0.790 | 98.16 |
| $[\lambda_1, \lambda_L] = [0.3, 0.5]$ | 0.318 | 0.802 | 94.30 |
| $[\lambda_1, \lambda_L] = [0.3, 0.6]$ | 0.309 | 0.811 | 90.92 |
| $[\lambda_1, \lambda_L] = [0.4, 0.5]$ | 0.310 | 0.808 | 95.73 |

generation quality for fine-grained interactions and semantic consistency. This is mainly because the iterative paradigm offers a global view of the entire story, thus reducing error accumulation and alleviating the propagation of the reference image flaws.

In Tab. 5 and Fig. 8, we validate our linear weighting strategy by comparing it with the fixed weighting strategy. While a smaller fixed weight factor enhances early text-image alignment, it limits consistency in later stages. Conversely, a larger fixed weight factor enforces excessive consistency across iterations, reducing flexibility. Thus, the fixed weighting strategy proves to be suboptimal. Based on the ablation results in Tab. 5, we adopt a linear weighting strategy: $[\lambda_1, \lambda_L] = [0.3, 0.5]$ in 10 iterations. This enables greater flexibility within the iterative paradigm.

**Initialization.** To ablate the effect of the initialization, we use a sequence of images consisting of the characters as reference images (*i.e.*, *w/o* initialization). Adopting the same character as the reference image for the story sequence resulted in a high consistency of all story visualization results, triggering a better aFID. However, Tab. 4 shows that when removing the initialization, there is a significant decrease in the image-text alignment of Story-Iter in terms of CLIP-T. Fig. 10 illustrates that without initialization, the diffusion model fails to generate required objects according to texts.

## 5 CONCLUSIONS AND DISCUSSIONS

We introduce Story-Iter, a training-free *iterative paradigm* for long story visualization. By using full-length generated images from the previous external iteration as reference images, our framework maintains semantic consistency and enhances generative quality for fine-grained interactions throughout the story, effectively reducing error accumulation and avoiding the propagation of flaws. Extensive experiments demonstrate that Story-Iter outperforms existing methods on the regular-length story visualization dataset, and shows strong results in long story visualization. These indicate the potential of the proposed iterative paradigm to advance long story visualization.

## 6 ETHICS STATEMENT

All authors of this work have read and commit to adhering to the ICLR Code of Ethics.

## 7 REPRODUCIBILITY

To ensure reproducibility, we provide pseudocode in Algo. 1 and implementation details in Sec. A.4. The full code can be found in the *Supplementary Material*.

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

TECHNICAL APPENDICES

## A PARADIGMS

Existing story visualization methods usually employ the Auto-Regressive (AR) or Reference-Image (RI) paradigms. In this work, we propose a novel iterative paradigm for story visualization. Next, we will discuss different story visualization paradigms in detail.

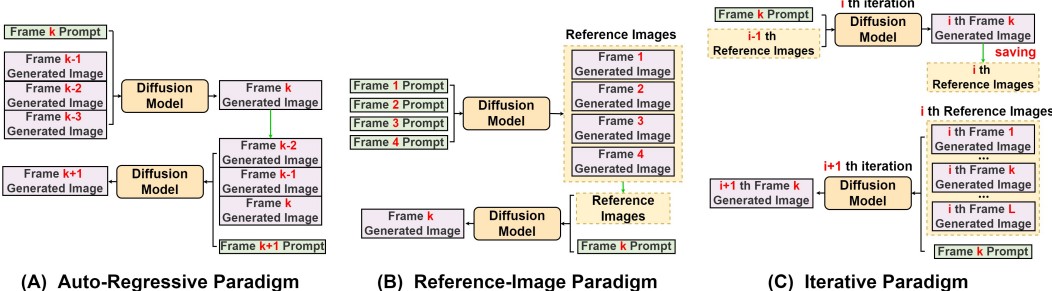

Figure 11: Different paradigms for story visualization. Zoom in for a better view.

### A.1 AUTO-REGRESSIVE PARADIGM

**Setting.** As shown in Fig. 11, AR paradigm-based methods typically use a limited number of previous frames and the corresponding text prompt of the current frame to guide current image generation. This helps the methods maintain semantic consistency between consecutive frames.

**Discussion.** However, the AR paradigm cannot consider future frames when synthesizing the current image, which makes the AR paradigm only maintain semantic consistency in neighboring frames but not throughout the story. Besides, the AR paradigm easily suffers from error accumulation. Therefore, the image quality of the AR paradigm gets worse as the length of the story increases.

### A.2 REFERENCE-IMAGE PARADIGM

**Setting.** RI paradigm-based methods employ the beginning visualized frames as reference images to guide the visualization of the rest of the story when performing long story visualization (see Fig. 11). Bootstrapping based on fixed reference images helps the methods to effectively maintain identity consistency in long story visualizations.

**Discussion.** However, such a setup ignores the consistency of emerging characters in the story, and all visualizations are affected by flaws in the reference images. Both issues affect the quality of long story visualizations with the RI paradigm.

### A.3 ITERATIVE PARADIGM

**Setting.** To address the aforementioned limitations, we propose an iterative paradigm in Story-Iter (Fig. 11). We constantly consider all generated images in the previous iteration with an iterative mechanism and model on the global embeddings. Specifically, when generating for the $k_{th}$ image, we propose to implement Global Reference Cross-Attention (GRCA) on global embeddings from all generated images in the previous iteration.

**Discussion.** By using all generated images from the previous iteration as reference images to guide the current generation, we effectively maintain semantic consistency throughout the story. Moreover, all the generated images as references are updated through each iteration. We illustrate in Fig. 12 how different reference images contribute to the generation of each image within the iterative paradigm. Taken together, the iterative paradigm effectively avoids the influence of defects in some reference images. We demonstrate the procedure of Story-Iter in Algo. 1.

**Algorithm 1** Pseudo-Code of Story-Iter.

```
# diffusion model:θ, iteration epochs:L, starting weight factor:λs,
    ending weight factor:λe, ith iteration jth diffusion step kth
    intermediate denoising features:I^i_{k,j}, story length:B, diffusion steps
    :J, decoder:D
# Initialize I^0_{k,j}, I^i_{k,j}~N(0, I), k~(1, B), i~(1, L), j~(0, J)
# Initialize Story-Iter iteration
for j in reversed(range(0, J)):
    # Init z~N(0, I) if j>1 else z=0
    I^0_{k,j-1}=(1/sqrt(αj))*I^0_{k,j}-(1-αj)*θ(I^0_{k,j},j,Tk)/sqrt(1-αj))+σt*z
R=concat([x^0_1,...,x^0_k,...,x^0_B]),  x^0_k=D(I^0_{k,0})

# Insert GRCA to θ and initialize weighting factor list λlist
λlist=linspace(λs,λe,L)
# Story-Iter Iteration
for i,λ in enumerate(λlist):
    for j in reversed(range(0, J)):
        I^i_{k,j-1}=(1/sqrt(αj))*(I^i_{k,j}-(1-αj)*θ(I^i_{k,j},j,Tk,R,λ)/sqrt(1-αj))+σt*z
        R=concat([x^i_1,...,x^i_k,...,x^i_B]),  x^i_k=D(I^i_{k,0})
```

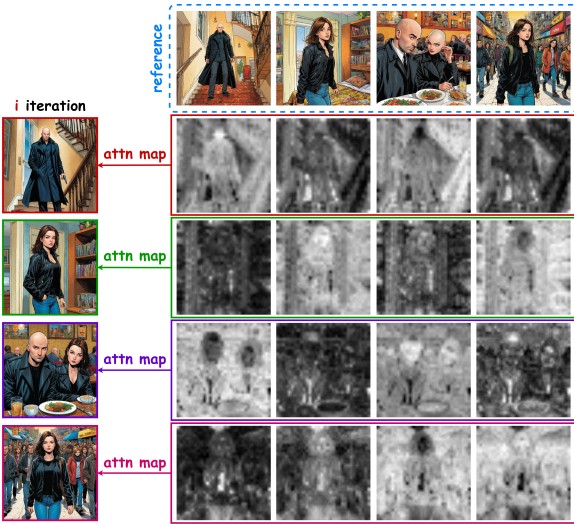

Figure 12: Attention maps for each reference image in the ith iteration of Story-Iter for each image generation.

Table 6: GRCA vs. CSA.

| Setting | CLIP-T ↑ | aCCS ↑ | aFID ↓ |
|---|---|---|---|
| GRCA | 0.322 | 0.757 | 105.17 |
| CSA | 0.315 | 0.768 | 102.44 |

| Setting | FLOPs ↓ | Time ↓ | VRAM ↓ |
|---|---|---|---|
| GRCA | 44.09T | 15S | 19GB |
| CSA | 22.32P | 19S | 40GB |

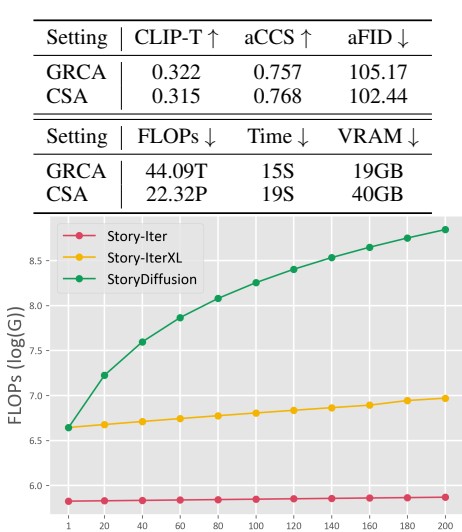

Figure 13: Computational feasibility with scaling reference images.

## A.4 IMPLEMENTATION DETAILS

To ensure a fair comparison, we used the weights of IP-Adapter (Ye et al., 2023) and IP-AdapterXL (Ye et al., 2023), respectively, resulting in two models: **Story-Iter** and **Story-IterXL**. We utilized DDIM (Song et al., 2020) for 50-step sampling with a Classifier-Free Guidance score set to 7.5. For the hyperparameters in our iterative paradigm, we set the number of story iterations to 10 by default. The weight factor $\lambda$ is set to 0.3 for the initial iteration and 0.5 for the final iteration, with linearly interpolated values for the intermediate iterations by our linear weighting strategy. Story-Iter is implemented in the NVIDIA RTX A6000.

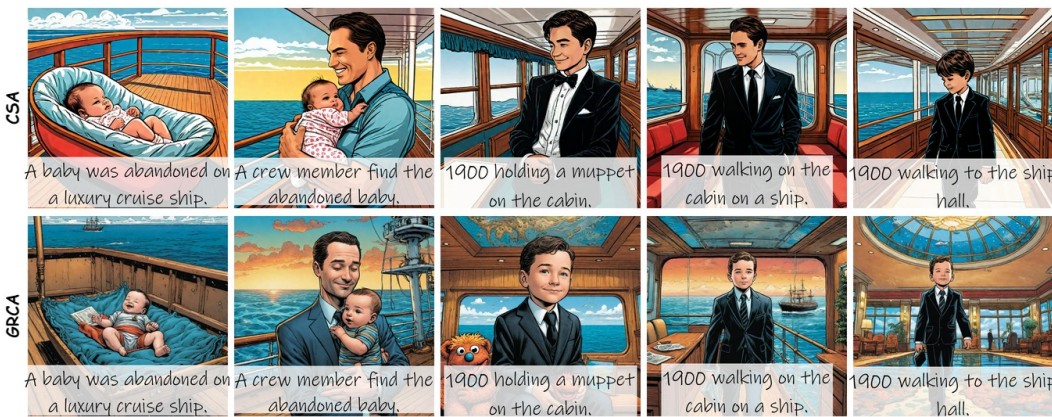

Figure 14: Qualitative comparison of GRCA and CSA.

## A.5 GRCA VS CSA

We investigate GRCA and CSA in Tab. 6 and Fig. 14, using the outputs of the first iteration from Story-Iter and StoryDiffusion, respectively. Though GRCA generates less visual consistency during the first iteration than CSA in terms of aCCS and aFID in Tab. 6, GRCA's global comprehension improves the consistency of multiple characters throughout stories shown in Fig. 14. For example, GRCA effectively preserves the consistency of emerging characters (*e.g.*, "*the character 1900*"), while CSA fails.

Establishing global story comprehension is fundamental to long-story visualization. Although the CSA can establish connections among all generated images during the generation process, its scalability for reference image scaling is limited, particularly for long stories, due to the high memory consumption of latent denoising representations. In Fig. 13, we analyze reference scaling of GRCA in single iteration and CSA in StoryDiffusion, without adopting the RI paradigm. FLOPs are calculated within the diffusion UNet. As shown in Fig. 13, as the number of reference images increases, StoryDiffusion experiences a significant rise in computation in terms of FLOPs, eventually becoming computationally prohibitive and can't maintain global story comprehension of long narratives. While Story-Iter and Story-IterXL are slightly affected. This lays the foundation for GRCA to preserve global story semantics in long story visualizations.

Though our focus is on addressing the fundamental modeling challenges in long story visualization, rather than optimizing for computational efficiency, we still discuss the computational cost of 50 diffusion steps for 1024-resolution image generation with 100 reference frames in Tab. 6. Since the Story-Iter paradigm involves multiple iterations, generating a full 100-frame story incurs high computational complexity (100 frames of 1024-resolution story visualization externally iterates one round at a cost of 4.30 PFLOPs/25 minutes). With that said, it is still easily scalable across a wide range of hardware devices since the generation or update of a single frame, the computational cost of 50 diffusion steps is **only 44.09 TFLOPs and takes 15 seconds** with 100 reference frames. The required VRAM is just 19 GB at 1024x1024 resolution. In comparison, for 1024x1024 resolution story visualization, StoryDiffusion generates 22.32 PFLOPS and occupies 31minutes. For each generated image, VRAM consumption is 40 GB, and inference time is 19 seconds. Moreover, with the introduction of global embedding in GRCA, Story-Iter can be easily expanded to 200 frames of story visualization. The actual computational limit mainly depends on the VRAM, e.g., 100 fps story visualization VRAM is 19GB, 200 fps story visualization VRAM is 24GB. Additionally, with 200 reference frames, the image generation time of Story-Iter is 17 seconds. This indicates that increasing the number of frames has minimal impact on the memory consumption and inference time of Story-Iter. Also, we note that **these costs can be reduced(∼50×)** using MeanFlow (Geng et al., 2025), distilled diffusion models (Kim et al., 2023a;b), or latent consistency models (Luo et al., 2023). As shown in Tab. 11 and Fig. 19, Story-Iter achieves high-quality results within just 10 iterations for 100-frame stories. Depending on the length and complexity of the given story, the number of external iterations in Story-Iter can be reduced accordingly, thereby further improving efficiency while still achieving high-quality story visualization. With these improvements, Story-Iter requires

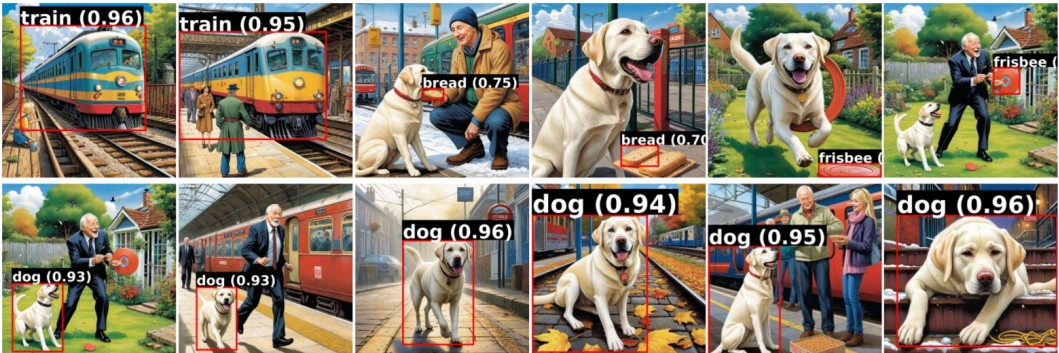

Figure 15: Calculation of aCCS and aFID.

only about **5 minutes** per external iteration to generate a 100-frame story at 1024×1024 resolution. Therefore, efficiency is unlikely to pose a limitation for Story-Iter in long story visualization.

## A.6 GRCA SETTING ABLATION

To systematically evaluate how different configurations of GRCA affect the generation quality of Story-Iter, we conduct a series of ablation experiments by replacing GRCA with simpler global context aggregation settings (see Tab. 7). All variants are tested under the same iterative generation paradigm to isolate the contribution of long-range dependency modeling.

Table 7: Ablation study on GRCA settings.

| Setting | CLIP-T | aCCS | aFID |
|---|---|---|---|
| Sliding Window | 0.313 | 0.774 | 101.08 |
| Random Reference Set | 0.306 | 0.750 | 119.52 |
| Mean-pooled Global Embedding | 0.311 | 0.741 | 108.19 |
| Story-Iter | 0.318 | 0.802 | 94.30 |

**Sliding Window.** This variant aggregates only the most recent 10 frames, capturing short-range temporal dependencies but failing to model long-range narrative structure. When characters or objects reappear after long intervals, the model lacks sufficient global reference cues, leading to degraded long-story consistency.

**Random Reference Set.** Random sampling introduces unstable and conflicting contextual signals. Such inconsistency accumulates across iterations and is particularly harmful in multi-character narratives, where mismatched global cues result in incoherent visual continuity.

**Mean-pooled Global Embedding.** Averaging all global features compresses scene-specific and character-specific information into a single homogenized vector, causing significant loss of structural and semantic cues. As a result, the model struggles to preserve cross-scene consistency.

These results demonstrate that long-range, content-adaptive attention over the entire story sequence is essential, and simple aggregation methods cannot replace GRCA without significant quality degradation.

## B EVALUATION METRIC

We use Grounding-DINO (Liu et al., 2023) to extract object bounding boxes, using the name of each repeated object in the story as the grounding text. For each repeated object, the first frame in which it appears is treated as the reference frame. aCCS (Cheng et al., 2024b) is defined as the average CLIP-I similarity between the reference crop and its corresponding crops in subsequent frames. aFID (Cheng et al., 2024b) is defined as the FID between the reference crop and the distribution of later crops of the same object. Because repeated objects appear across multiple frames and all reference crops are obtained from text-conditioned generation, these metrics are stable and not overly sensitive to detection noise (see Fig. 15).

Table 8: Human evaluation comparison of subject-consistent image generation, regular-length story visualization, and long story visualization. The best is highlighted in red.

| Subject-Consistent Image Generation | | | | |
|---|---|---|---|---|
| Model | Align. ↑ | Inter. ↑ | Cons. ↑ | Pref. ↑ |
| IP-Adapter (Ye et al., 2023) | 2.51 | 3.27 | 4.58 | 4.19 |
| IP-AdapterXL (Ye et al., 2023) | 2.66 | 3.36 | 4.72 | 4.26 |
| PhotoMaker (Li et al., 2024b) | 3.79 | 4.18 | 4.25 | 4.11 |
| StoryDiffusion (Zhou et al., 2024) | 4.15 | 4.28 | 4.50 | 4.48 |
| Story-Iter | 4.02 | 4.20 | 4.41 | 4.33 |
| Story-IterXL | 4.20 | 4.35 | 4.58 | 4.54 |
| Regular-Length Story Visualization | | | | |
| SDM (Rombach et al., 2022) | 4.11 | 2.37 | 2.01 | 1.14 |
| Prompt-SDM (Rombach et al., 2022) | 4.03 | 3.49 | 1.99 | 1.26 |
| Finetuned-SDM (Rombach et al., 2022) | 3.35 | 3.82 | 2.15 | 1.60 |
| AR-LDM (Pan et al., 2024) | 3.08 | 3.64 | 2.90 | 2.05 |
| StoryGen (Liu et al., 2024) | 3.72 | 4.17 | 3.83 | 3.39 |
| StoryDiffusion (Zhou et al., 2024) | 3.96 | 4.48 | 4.52 | 4.37 |
| Story-Iter | 3.89 | 4.21 | 4.36 | 4.10 |
| Story-IterXL | 4.06 | 4.60 | 4.74 | 4.62 |
| Long Story Visualization | | | | |
| AR-LDM (Pan et al., 2024) | 3.30 | 3.68 | 3.42 | 3.27 |
| StoryGen (Liu et al., 2024) | 3.51 | 4.06 | 3.88 | 3.51 |
| IP-Adapter (Ye et al., 2023) | 3.79 | 4.27 | 4.30 | 4.06 |
| IP-AdapterXL (Ye et al., 2023) | 3.83 | 4.23 | 4.61 | 4.11 |
| StoryDiffusion (Zhou et al., 2024) | 4.16 | 4.30 | 4.53 | 4.35 |
| Story-Iter | 3.97 | 4.15 | 4.42 | 4.29 |
| Story-Iter | 4.35 | 4.47 | 4.70 | 4.65 |

Table 9: Efficiency comparison with editing models for *100* frames *1024×* story visualization.

| Method | Diffusion Steps | External Iterations | FLOPs | VRAM | Time | CLIP-T | aCCS | aFID |
|---|---|---|---|---|---|---|---|---|
| Nano Banana | - | 1 | - | - | 33 min | 0.310 | 0.791 | 106.14 |
| FLUX.1.Kontext (Batifol et al., 2025) | 50 | 1 | 371 PFLOPs | 68GB | 48 min | 0.290 | 0.783 | 78.03 |
| Story-IterXL | 50 | 10 | 43 PFLOPs | 19GB | 250 min | 0.318 | 0.802 | 94.30 |
| Story-IterXL-Fast | 4 | 10 | 3 PFLOPs | 19GB | 20 min | 0.309 | 0.788 | 109.13 |

## C  HUMAN EVALUATION

**Setting.** To complement the evaluation metrics to accurately reflect the quality of the generated stories, we involve human evaluation to further compare Story-Iter and baselines. Referring to the setting in StoryGen (Liu et al., 2024), we invite participants to rate various aspects: text-image alignment (Align.), character interaction (Inter.), content consistency (Cons.), and preference (Pref.) on a scale from 1 to 5. The higher the better.

**Results.** Tab. 8 shows that our Story-Iter receives more preference from the participants. It is worth noting that although IP-Adapter receives higher scores for consistency in the subject-consistent image generation task, Story-Iter is more favored in text-image alignment and generating character interactions. For regular-length and long story visualization, Story-Iter is more preferred compared to baselines in most evaluation aspects, especially visual consistency and capability to generate character interactions. This is aligned with the quantitative measurement.

## D  COMPARISON WITH EDITING MODELS

In this section, we provide a further comparative analysis between Story-Iter and the latest reference-image based image editing models.

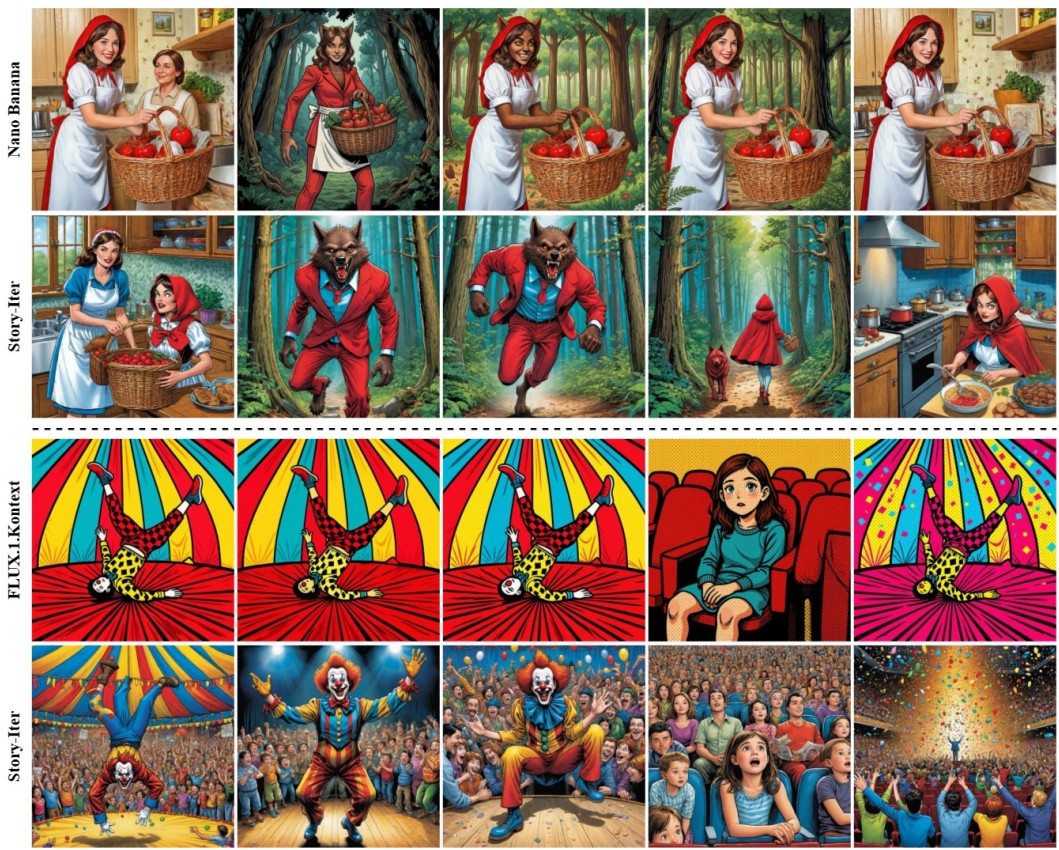

Figure 16: Overly rigid identity constraints for Reference-image based image editing models.

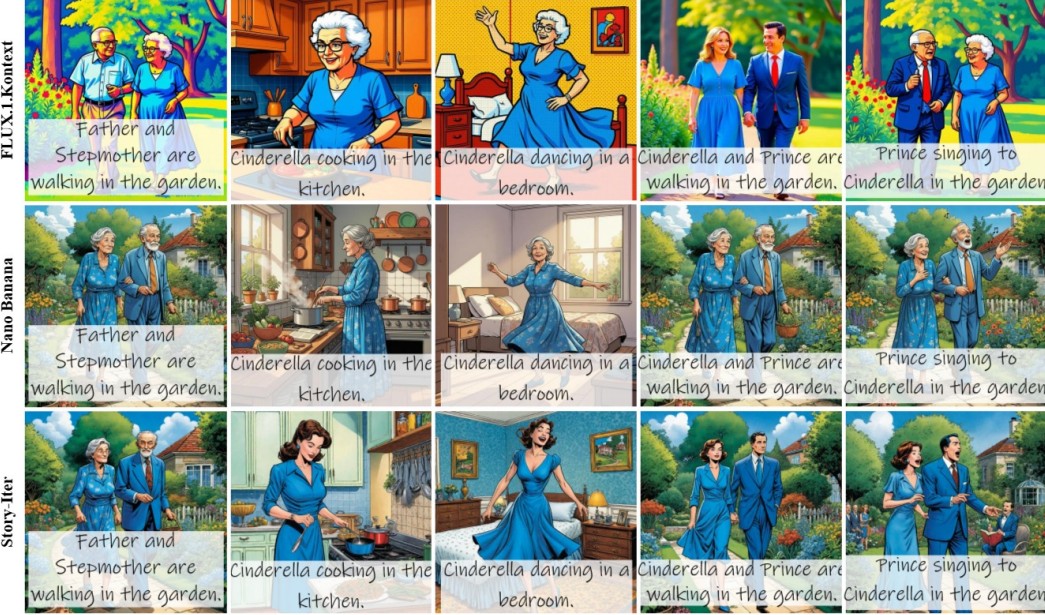

Figure 17: New character missing for Reference-image based image editing models.

**Qualitative Evaluation.** As shown in Tab. 9, Fig. 16 and Fig. 17, although reference-image based editing models can help maintain object consistency in story visualization, they impose overly rigid identity constraints compared with Story-Iter. Models such as Flux.1.Kontext (Batifol et al., 2025), and Nano-Banana are designed to preserve the appearance of the reference image as strictly as possi-

ble. While this behavior is desirable for image editing, it introduces a critical limitation in long-story visualization: the character remains nearly identical across frames, regardless of changes in action, pose, or narrative context. This severely reduces intra-story diversity and leads to lower CLIP-T scores. Reference-image based editing models also struggle to handle new characters or objects that do not appear in the reference image. Long stories frequently introduce new entities, for which models like Flux.1.Kontext and Nano-Banana cannot maintain cross-frame consistency. This results in degraded CLIP-T performance and unstable behavior in multi-character scenarios. This limitation is reflected in the lower CLIP-T of Flux.1.Kontext, as well as its difficulty in maintaining consistency when multiple characters are present. In contrast, Story-Iter is designed for open-ended, multi-character narratives rather than single-image editing. It employs a training-free, content-adaptive GRCA mechanism that maintains consistency across diverse scenes and newly introduced entities, without relying on a fixed reference image.

**Efficiency Comparison.** Furthermore, Tab. 9 presents an additional efficiency comparison between Story-Iter and the reference-image based editing models. As shown in Tab. 9, Story-Iter is substantially more efficient than reference-image based editing models such as Nano Banana and FLUX.1.Kontext when generating long stories. In particular, Story-IterL requires only 43 PFLOPs and 19GB VRAM for a 100-frame sequence across 10 external iterations, while FLUX.1.Kontext incurs 371 PFLOPs and 68GB VRAM even for a single pass, making the editing-based models significantly more expensive despite lacking multi-iteration refinement. However, despite its efficiency advantage over editing models, Story-Iter still exhibits high inference time (250 minutes for 100 frames), due to the external iterations. To address this limitation, we introduce Story-Iter-Fast, a variant built upon the SDXL-LCM backbone. By reducing diffusion sampling from 50 steps to only 4, this fast version improves efficiency dramatically—achieving a $12\times$ speed-up (250 min to 20 min) and a $14\times$ reduction in FLOPs (43 PFLOPs to 3 PFLOPs)-while maintaining comparable story-level consistency to the Story-Iter. This demonstrates that our iterative GRCA framework remains effective even under highly accelerated diffusion settings.

## E   SUBJECT-CONSISTENT GENERATION COMPARISON

**Related Work.** Subject consistency is critical for tasks such as story visualization and video generation. Recent advancements in subject-consistent image generation have focused on reducing computation while maintaining consistency. Early approaches like Gal et al. (2022); Ruiz et al. (2023) require extensive fine-tuning, prompting more efficient methods (Ryu, 2023; Han et al., 2023; Kumari et al., 2023; Yuan et al., 2023; Tewel et al., 2024; Cao et al., 2023; Xiao et al., 2024). Notable progress (Li et al., 2024b; Avrahami et al., 2024; Ye et al., 2023; Li et al., 2024a; Wei et al., 2023; Xu et al., 2024) includes IP-Adapter (Ye et al., 2023) with its decoupled cross-attention module. However, IP-Adapter is a single image generation model, which cannot be directly applied to story visualization that requires establishing connections for all frames.

In the evaluation phase, we employ GPT-4o (OpenAI, 2024) according to the settings of StoryDiffusion (Zhou et al., 2024) to generate 20 character descriptions and 100 specific activity descriptions, respectively. We combine them as 2000 test descriptions, to compare Story-Iter and subject-consistent image generation baselines, including IP-Adapter (Ye et al., 2023), PhotoMaker (Li et al., 2024b), and StoryDiffusion (Zhou et al., 2024).

**Quantitative Evaluation.** For quantitative comparisons on subject-consistent image generation, we employ CLIP text-to-image similarity (CLIP-T) and image-image similarity (CLIP-I) to measure consistency between the character images and generated images. Tab. 10 shows that Story-Iter achieves SoTA performance in terms of both quantitative metrics, which demonstrates Story-Iter's ability to generate subject-consistent image sequences based on text prompts or image prompts.

**Qualitative Evaluation.** Fig. 18 shows the qualitative comparison results. Story-Iter generates higher-quality images in subject-consistent and detailed interactions. In contrast, IP-Adapter fails to generate correctly, *e.g.*, "*paper*", "*whiteboard*", and "*chainsaw*". PhotoMaker cannot generate images consistently, *e.g.*, maintaining details of the attire. Despite accurately generating content according to text prompts with visual consistency, StoryDiffusion suffers from visualizing complex details due to lacking global story comprehension. By incorporating a global story view in our

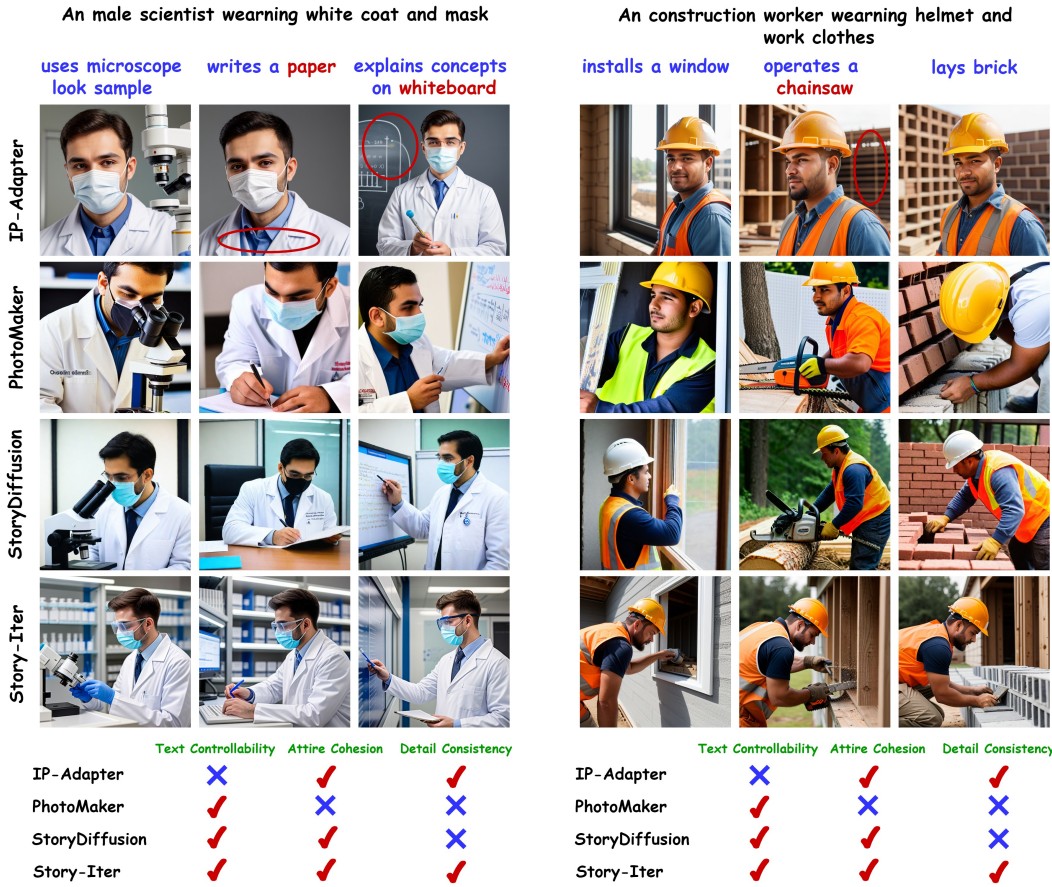

Figure 18: Qualitative comparison of subject-consistent image generation methods.

Table 10: Quantitative comparison in subject-consistent image generation tasks.

| Method | CLIP-T ↑ | CLIP-I ↑ |
|---|---|---|
| IP-Adapter (Ye et al., 2023) | 0.307 | 0.872 |
| Story-Iter (Ours) | 0.326 | 0.877 |
| IP-AdapterXL (Ye et al., 2023) | 0.312 | 0.879 |
| PhotoMaker (Li et al., 2024b) | 0.317 | 0.880 |
| StoryDiffusion (Zhou et al., 2024) | 0.330 | 0.882 |
| Story-IterXL (Ours) | **0.332** | **0.884** |

Table 11: Quantitative comparison of multiple iterations.

| Iteration | CLIP-T ↑ | aCCS ↑ | aFID ↑ |
|---|---|---|---|
| initialization | 0.330 | 0.502 | 214.94 |
| $1_{th}$ iteration | 0.322 | 0.757 | 105.17 |
| $3_{th}$ iteration | 0.325 | 0.770 | 110.86 |
| $5_{th}$ iteration | 0.319 | 0.783 | 100.81 |
| $10_{th}$ iteration | 0.306 | 0.840 | 91.35 |
| $15_{th}$ iteration | 0.297 | 0.848 | 90.62 |

iterative paradigm, Story-Iter can maintain visual consistency, especially in details throughout the story.

## F  MORE ITERATIONS

**Setting.** In this section, we compare results on different iterations in the iterative paradigm and investigate the impact of longer iterations on story visualization. Specifically, we study the visualization results in the initialization, $1_{st}$, $5_{th}$, $10_{th}$, and $15_{th}$ iterations, respectively.

**Results.** Tab. 11 shows that as iteration increases, Story-Iter achieves significant improvement in visual consistency (aCCs and aFID) while text-image alignment (CLIP-T) drops slightly. This further demonstrates the contribution of the iterative paradigm to the semantic consistency of the overall story. However, we also note that a further increase in iterations harms text-image alignment,

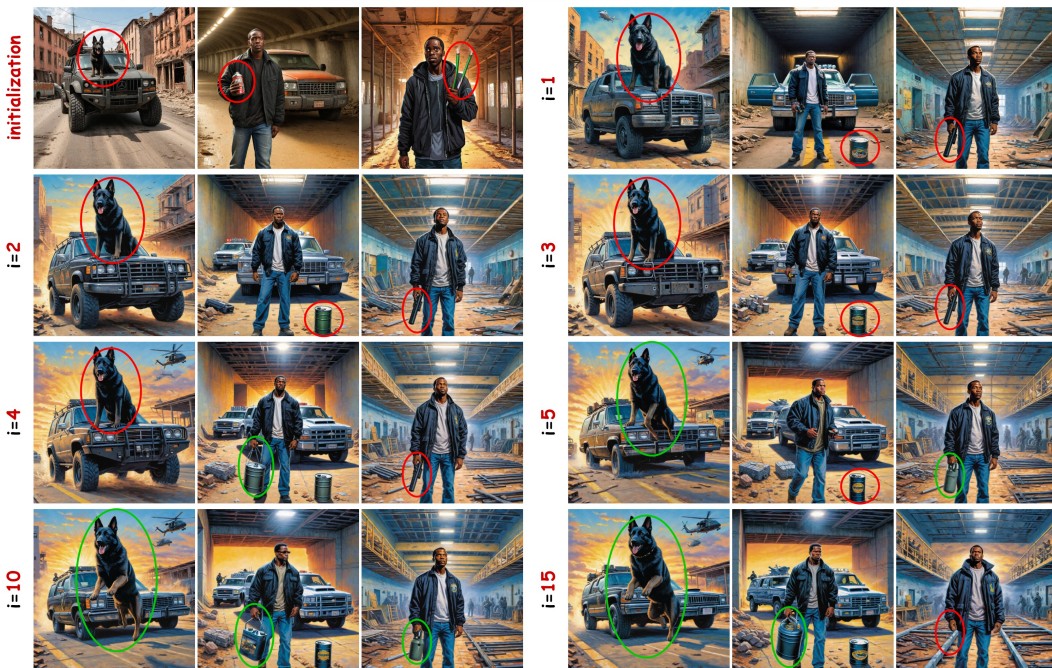

Figure 19: Story visualization results from different iterations by Story-Iter. Accurate interactions are denoted in green, wrong or missing ones are in red.

with limited gain in visual content consistency. This indicates that while Global Reference Cross-Attention (GRCA) effectively improves the content consistency of the long story, the increasing weighting factor of GRCA during the iterations poses a challenge to aligning the text prompts.

Fig. 19 demonstrates a significant improvement in generative quality for fine-grained interactions as the iteration proceeds. The iterative paradigm effectively alleviates the diffusion model's limitations on complex interaction generation by continuously creating input channels for text prompts. But more iterations wouldn't improve the generation quality further. Therefore, based on the above results, fewer than 10 iterations are sufficient to achieve high-quality story visualization, with any further iterations yielding negligible improvements.

This trade-off between consistency and text-image alignment is analogous to autoregressive models, where increased sequence length may introduce compounding errors, yet also allows for stronger global coherence. This tradeoff does not indicate inefficiency in our iterative refinement process, but rather highlights the importance of balancing local fidelity and global consistency. Importantly, this trade-off can be effectively mitigated. Also, as demonstrated in Tab. 11, adjusting the hyperparameters based on the specific story length and content, as well as terminating the external iteration of Story-Iter earlier, can offer a better trade-off. This opens promising directions for future work, where we plan to integrate content-aware and length-sensitive hyperparameter scheduling, guided by automatic evaluations (e.g., GPT-based metrics), to further optimize both consistency and alignment.

## G LIMITATION

While Story-Iter demonstrates great potential in long-story visualization, there remains an inherent trade-off between consistency and diversity. Although our initialization and linear weighting strategy help maintain global diversity throughout the story, certain local frames still exhibit diversity

limitations. In future work, we aim to incorporate pose or layout conditional controls to further encourage diversity between consecutive frames.

At the same time, multiple external iterations for long story generation may affect the practical efficiency of Story-Iter. However, this limitation can be fundamentally addressed through accelerated generation methods such as MeanFlow (Geng et al., 2025), which we discuss in Sec. A.5.

Additionally, we observe a trade-off between iterations and text–image alignment: longer iterations enhance global consistency but weaken alignment. This does not indicate inefficiency of the iterative refinement process, but rather reflects the balance between local fidelity and global consistency. Notably, this trade-off can be mitigated by tuning hyperparameters or terminating iterations earlier, as discussed in Sec. F. For future work, we plan to introduce Qwen3-VL–based content-aware scheduling, where Qwen3-VL evaluates story semantics during refinement and adaptively decides: whether an iteration should stop early, and set the next iteration's GRCA/text-conditioned weighting. This content-adaptive controller will further optimize consistency–alignment trade-offs without sacrificing Story-Iter's strong narrative coherence. In addition, we plan to expand our benchmark and release a larger corpus to support more comprehensive and standardized evaluation for the community.

## H    MORE VISUALIZATION RESULTS

In this section, we provide more visualization results from Story-Iter and the baselines.

### H.1    VISUAL COMPARISON

We compare the long story visualization results of representative work with AR-based, RI-based, and iterative paradigms, respectively. Specifically, Fig. 24, Fig. 20, Fig. 21, Fig. 23, and Fig. 22 show the generated results of the same "*Robinson*" story from the proposed Story-Iter (iterative), StoryGen (Liu et al., 2024) (AR-based), IP-Adapter (Ye et al., 2023) (subject-consistent image generation), ConsiStory (Tewel et al., 2024) (RI-based), and StoryDiffusion (Zhou et al., 2024) (RI-based), respectively.

**Results.** Fig. 20 shows that the visualization quality from StoryGen constantly gets worse as the length of the story increases. The visualization results of the IP-Adapter in Fig. 21 show obvious interaction missing or wrong interactions. In Fig. 22, StoryDiffusion maintains high visual quality throughout the story but suffers from some shortcomings in multi-character interactions. In contrast, our Story-Iter effectively achieves high-quality story visualization and addresses the aforementioned limitations (see Fig. 24).

We also observe that, although Story-Iter employs global embeddings, it still preserves fine-grained consistency compared to methods relying heavily on local visual embeddings—such as StoryDiffusion, StoryGen, and ConsiStory. By summarizing an entire image into a compact representation, global embeddings enable computationally affordable processing while preserving subject relevance and semantic consistency across full-length story sequences. At the same time, our method leverages **latent denoising features** that retain pixel-wise, fine-grained visual information, guided by the global embedding. This combination helps preserve important local details, such as clothing and facial features, as illustrated in Fig. 24. In contrast, StoryDiffusion, StoryGen, and ConsiStory paradoxically yield inconsistent details, as they are substantially disrupted by irrelevant noise due to their excessive emphasis on local features.

### H.2    STORY-ITER-FAST

We provide qualitative long-story visualization result of Story-Iter-Fast in Fig. 25. This result demonstrates that the proposed GRCA mechanism and iterative framework are fully compatible with distilled and accelerated diffusion models; even under extremely efficient sampling regimes, Story-Iter-Fast is able to maintain strong global story coherence.

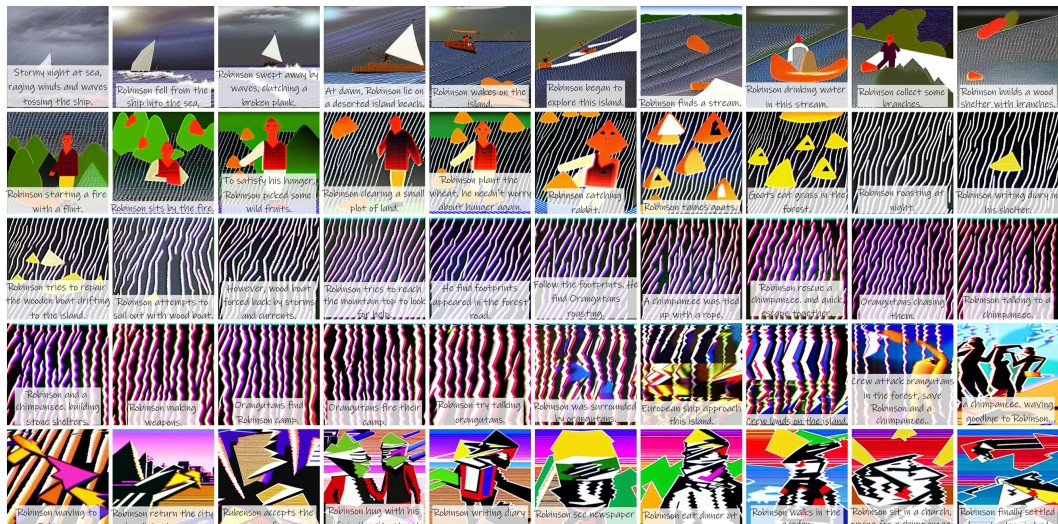

Figure 20: Visualization results of StoryGen for the "*Robinson*" story. Zoom in for a better view.

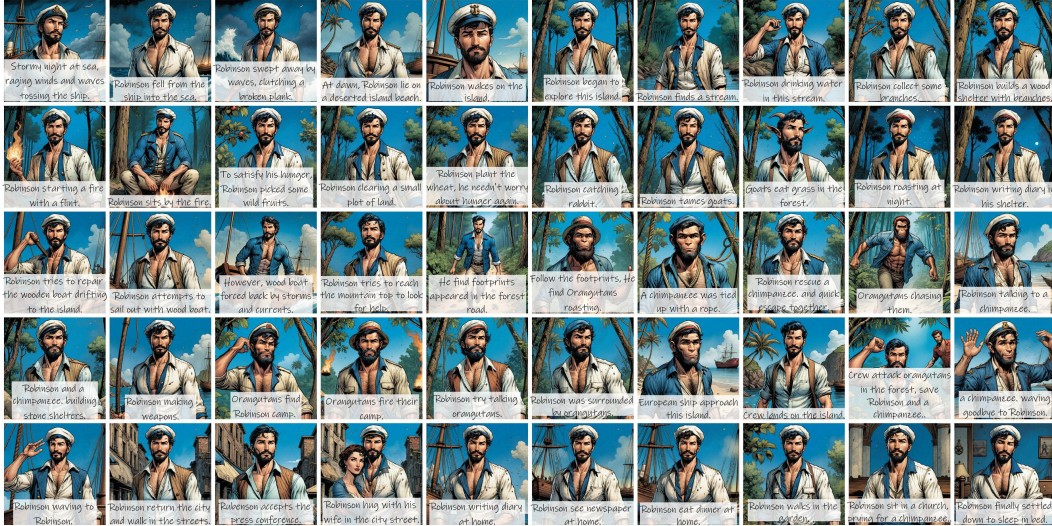

Figure 21: Visualization results of IP-Adapter for the "*Robinson*" story. Zoom in for a better view.

### H.3 DIFFERENT STYLE

We visualize scenes of the same long story in different styles in Fig. 26, Fig. 27, and Fig. 28. Meanwhile, we provide more long story visualization examples from Story-Iter in different styles in Fig. 29, Fig. 30, and Fig. 31. The experiment results suggest that Story-Iter can be applied to different visual styles as well. Furthermore, these results demonstrate that Story-Iter consistently generates high-quality visual narratives across a variety of initialization conditions—including different styles, content types, story lengths, and random seeds—highlighting its robustness and generalizability.

### H.4 LONGER STORY VISUALIZATION RESULTS

In Fig. 32 and Fig. 33, we show the visualization results of the long story (up to 100 frames).

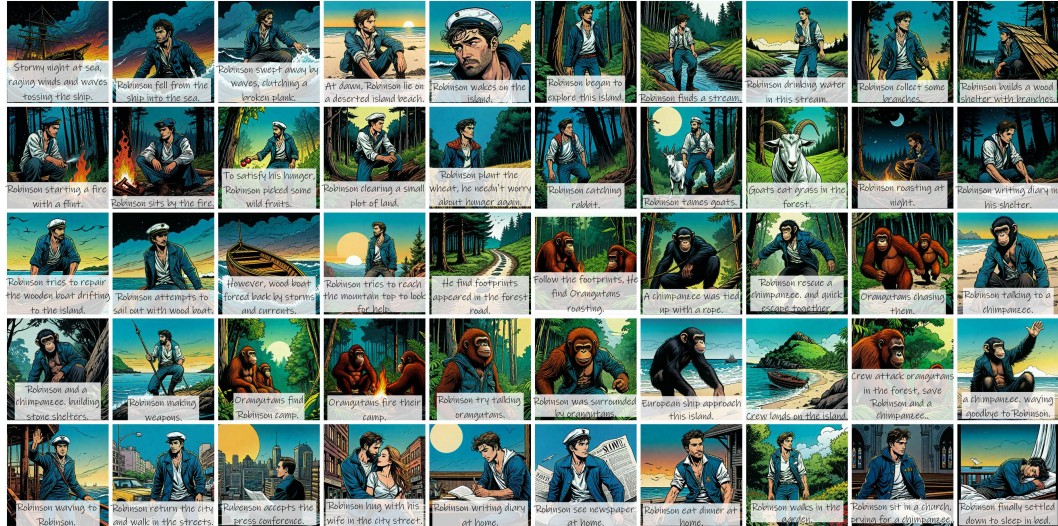

Figure 22: Visualization results of StoryDiffusion for the "*Robinson*" story. Zoom in for a better view.

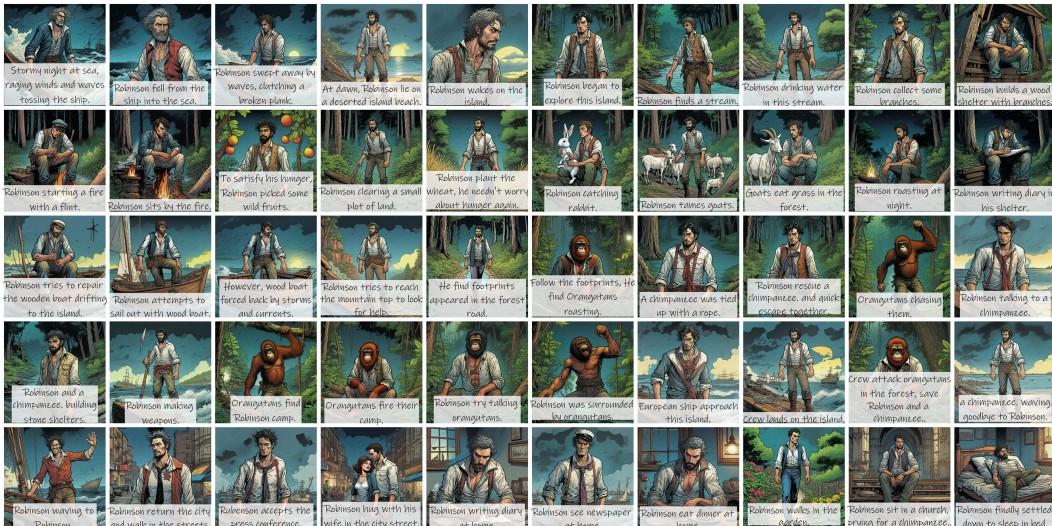

Figure 23: Visualization results of ConsiStory for the "*Robinson*" story. Zoom in for a better view.

# I DECLARATION OF LLM TOOL USAGE

During the preparation of this manuscript, I used OpenAI's GPT-4.1 model for minor language refinement and smoothing of the writing. The LLM tool was not used for generating original content, conducting data analysis, or formulating core scientific ideas. All conceptual development, experimentation, and interpretation were conducted independently without reliance on LLM tools. The other points involving the use of LLMs have already been highlighted in the paper.

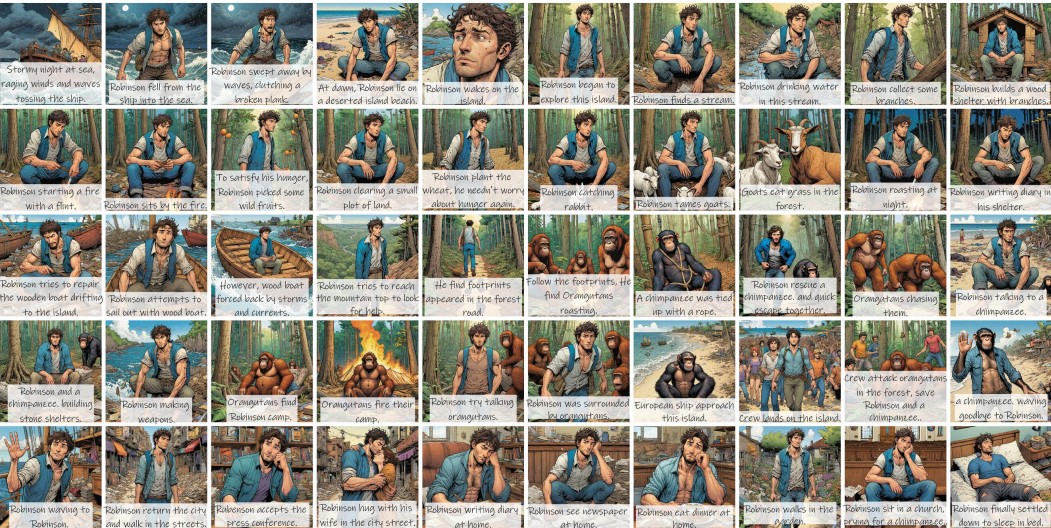

Figure 24: Visualization results of our Story-Iter for "*Robinson*". Zoom in for a better view.

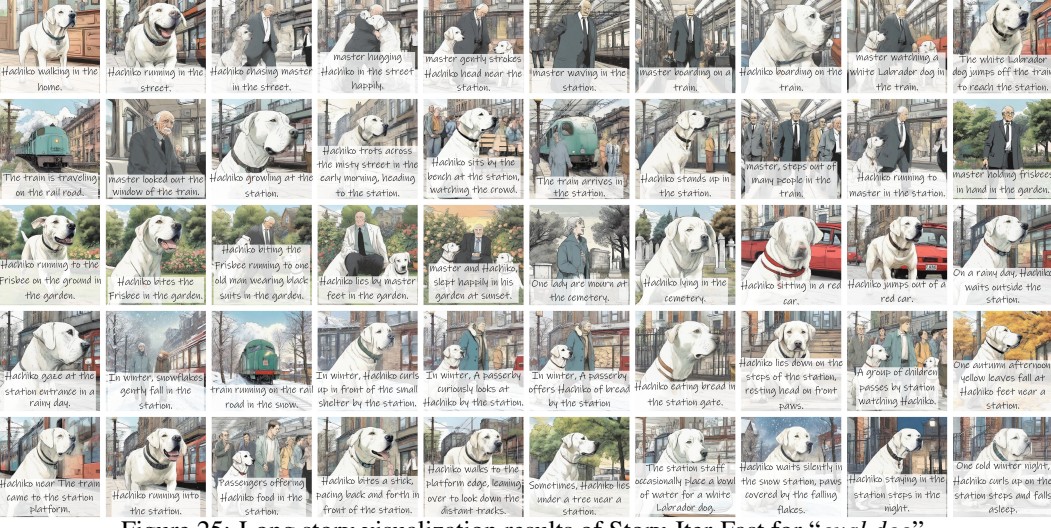

Figure 25: Long story visualization results of Story-Iter-Fast for "*oyal dog*".

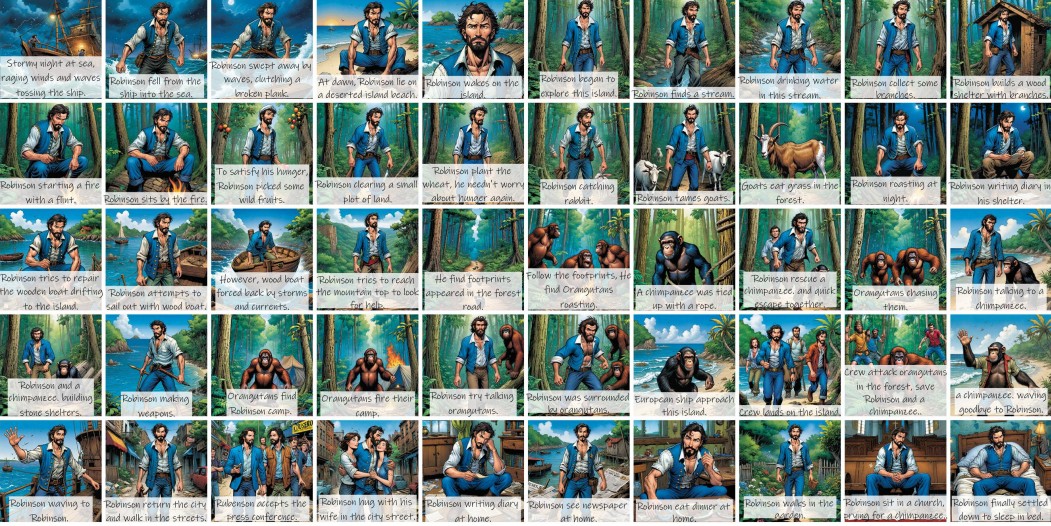

Figure 26: Our comic style story visualization results for "*Robinson*". Zoom in for a better view.

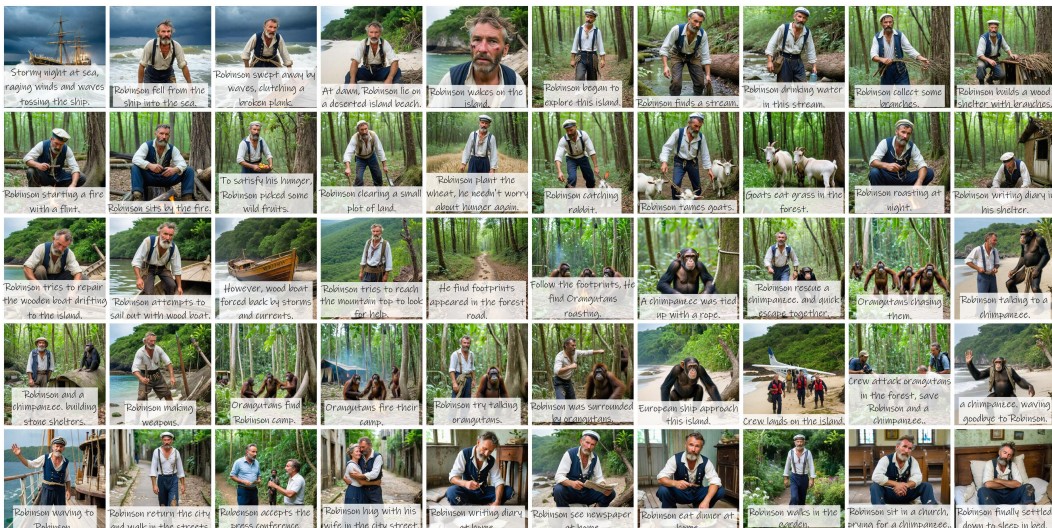

Figure 27: Our realistic style story visualization results for "*Robinson*". Zoom in for a better view.

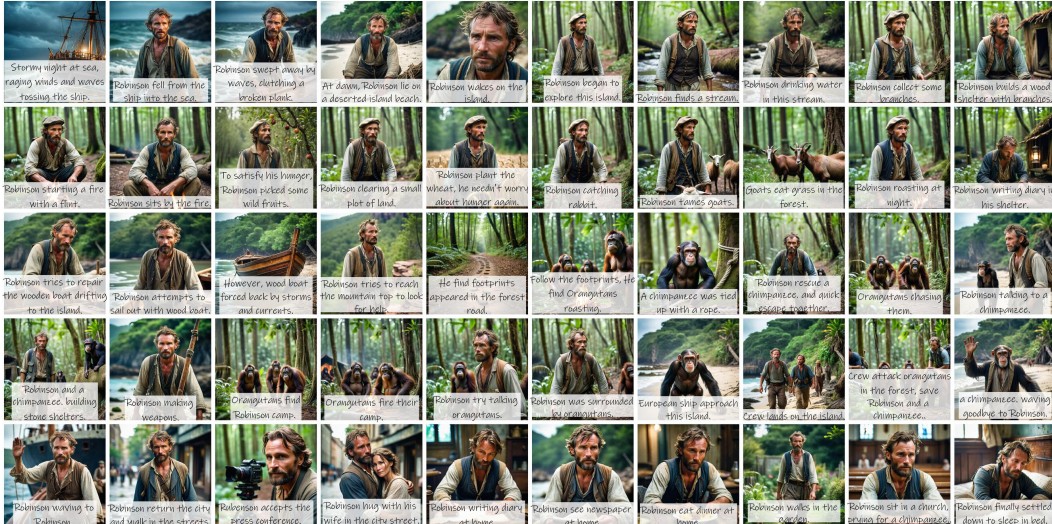

Figure 28: Our film style story visualization results for "*Robinson*". Zoom in for a better view.

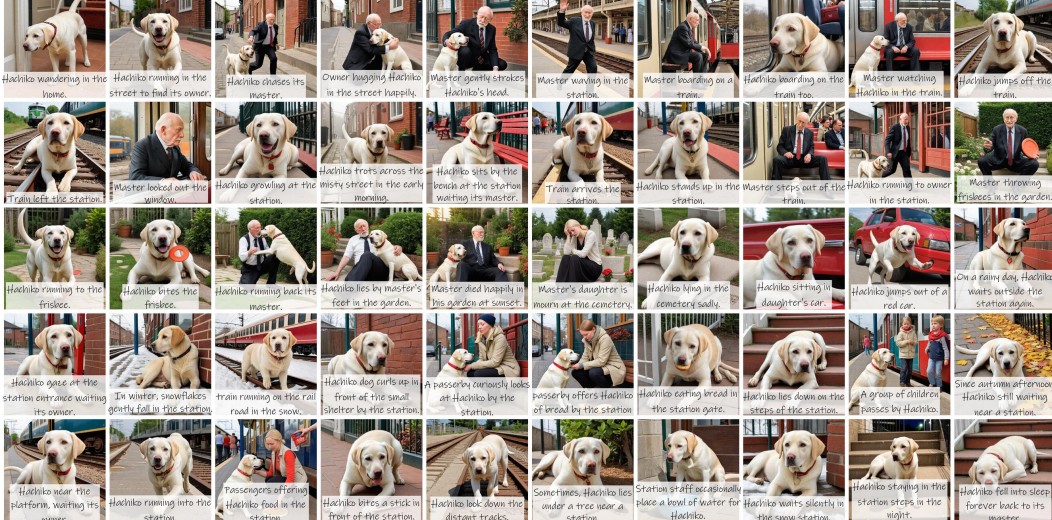

Figure 29: Our realistic style story visualization results for "*loyal dog*". Zoom in for a better view.

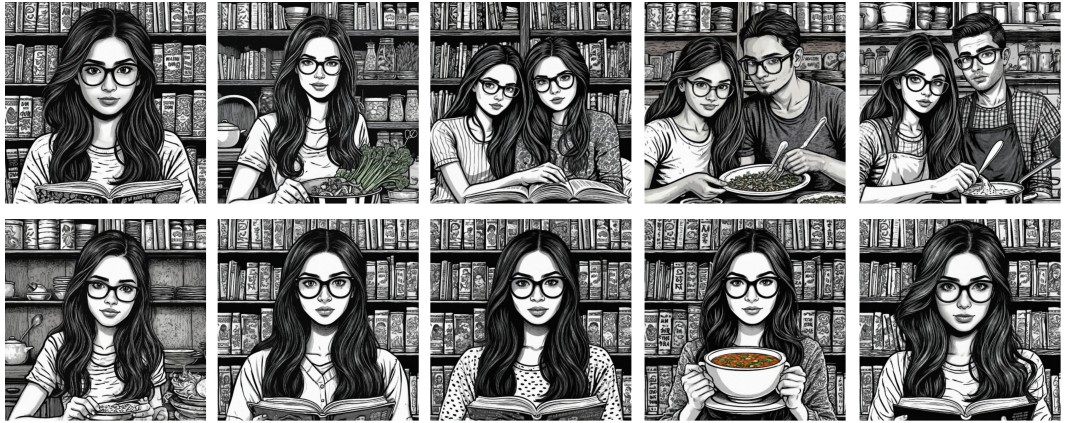

Figure 30: Our monochrome style regular story visualization results in StorySalon (Liu et al., 2024). Zoom in for a better view.

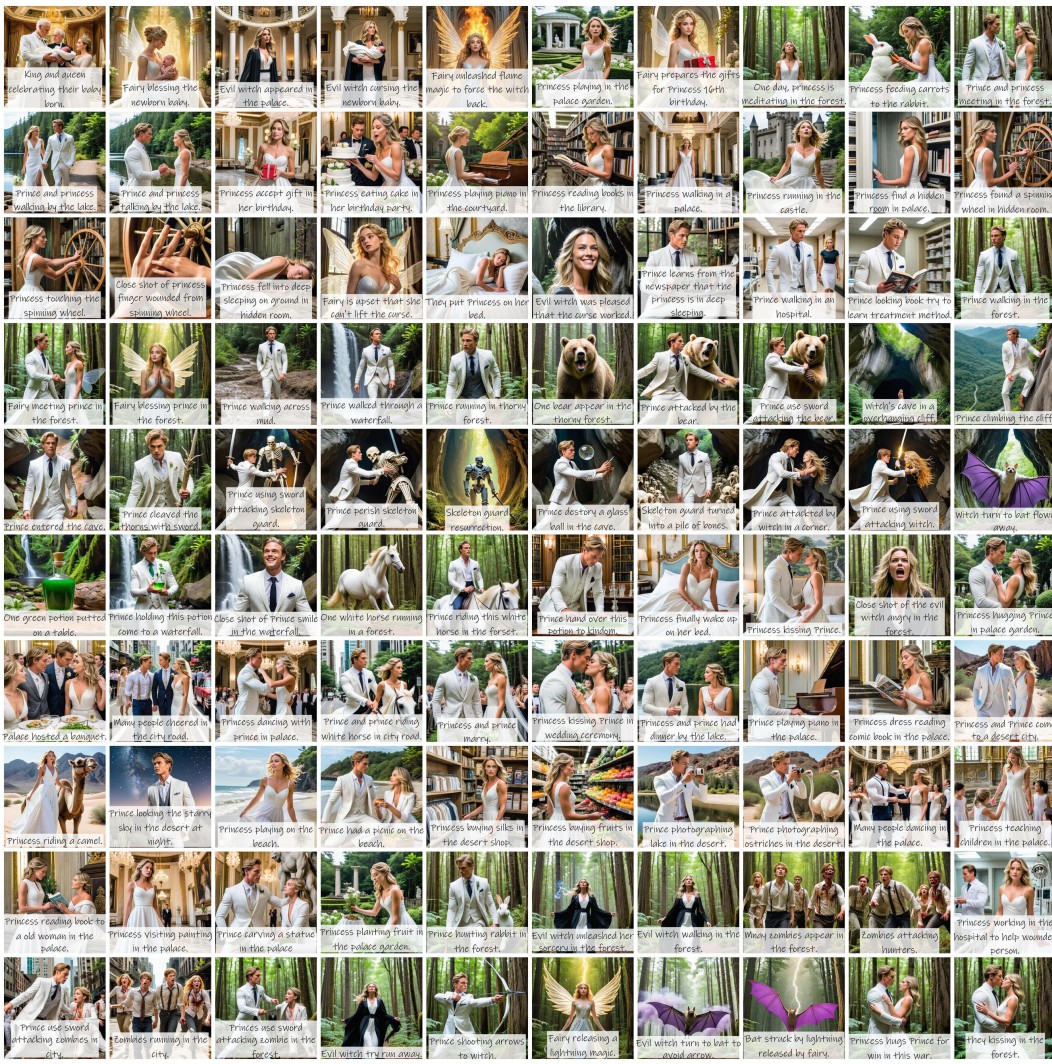

Figure 31: Our realistic style story visualization results for "*The Prince and the Princess*". Zoom in for a better view.

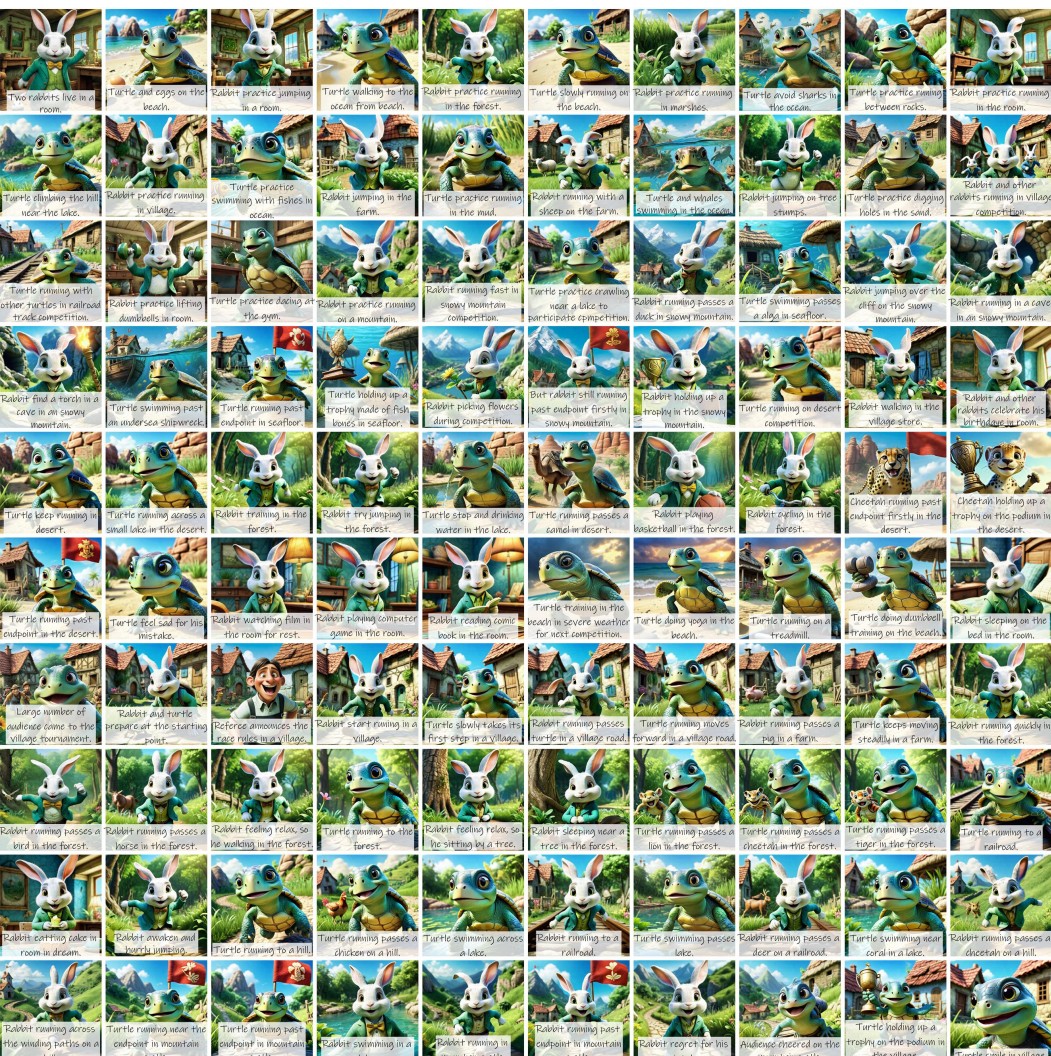

Figure 32: Our long story visualization results for "*Tortoise and the Hare race*". Zoom in for a better view.

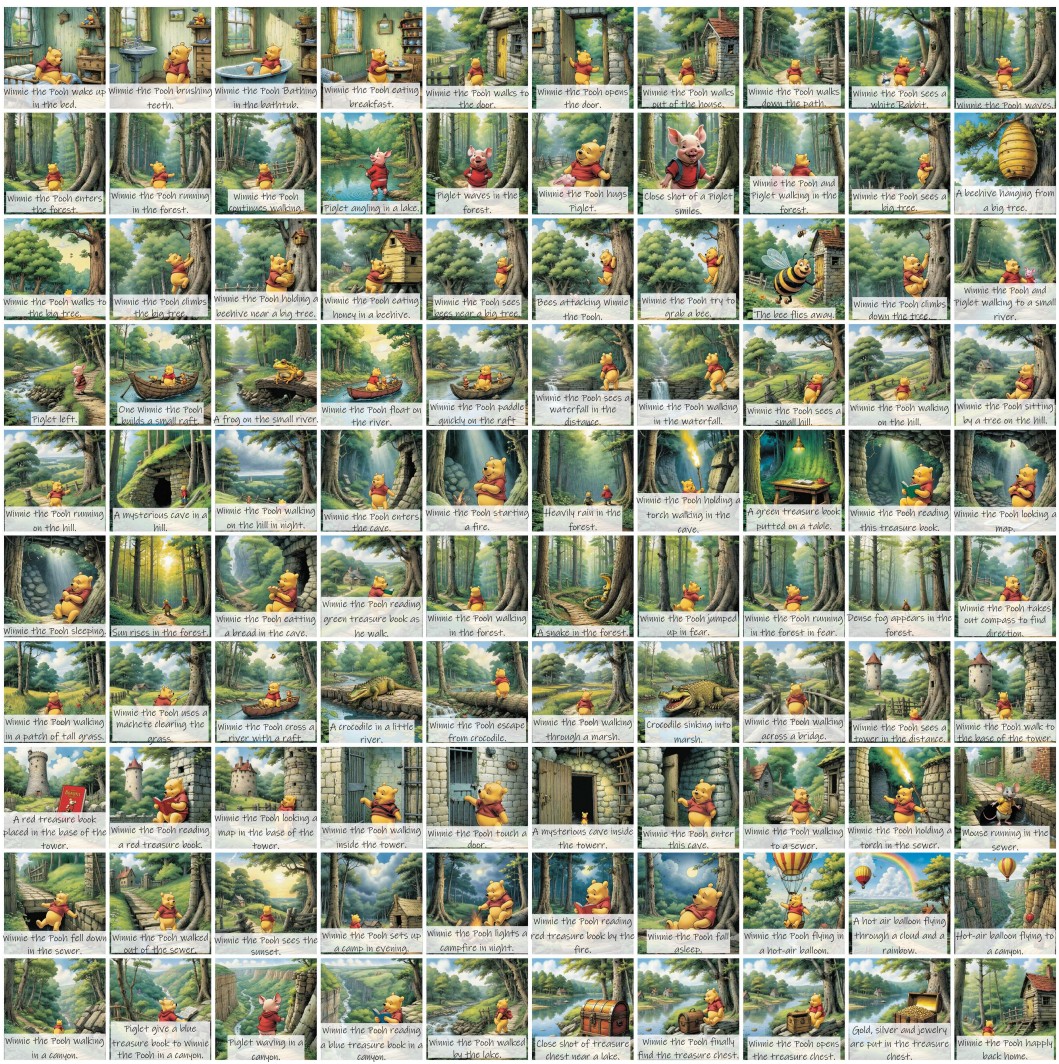

Figure 33: Our long story visualization results for "*Winnie the Pooh*". Zoom in for a better view.

