# OpenReview forum: "Story-Iter: A Training-free Iterative Paradigm for Long Story Visualization"
_ICLR.cc/2026/Conference — ICLR 2026 Poster_

### Official Review · Reviewer_LXWx · 2025-11-01

**Soundness:** 3
**Presentation:** 3
**Contribution:** 2
**Rating:** 4
**Confidence:** 4

**Summary:**

This paper introduces Story-Iter, a training-free iterative paradigm for long story visualization. Unlike existing autoregressive or fixed-reference methods, Story-Iter progressively refines each frame across external iterations by referencing all frames from the previous round. The key contribution is a Global Reference Cross-Attention (GRCA) module that uses compact global embeddings instead of high-dimensional latent features, enabling scalable processing of sequences up to 100 frames. A linear weighting strategy balances visual consistency and text alignment across iterations. Experiments show state-of-the-art results: over StoryGen and StoryDiffusion, with showing some good fine-grained interaction generation and detail preservation.

**Strengths:**

1) Novel iterative paradigm: External iterations that refine all frames by referencing the complete previous sequence, effectively addressing error accumulation in autoregressive methods and fixed-reference limitations
2) Scalable GRCA module: Uses compact global embeddings rather than high-dimensional latent features, enabling 100+ frame stories with manageable memory (19GB vs 40GB for StoryDiffusion)
3) Strong empirical validation: Consistent improvements across multiple metrics and benchmarks, with comprehensive ablations, human evaluation, and comparisons to diverse baselines
4) Training-free and practical: Plug-and-play approach reusing IP-Adapter weights, no dataset-specific training required
5) Thorough experimental design: Addresses both regular-length (StorySalon) and long stories (up to 100 frames), with qualitative evidence of improved fine-grained interactions

**Weaknesses:**

1) Base model (IP-Adapter) outdated. Even though we are in the era of Flux/SD3.5, use of IP-Adapter-based models with SD 1.5 (I am not sure whether it is 1.5 - I checked the code and you only provided StoryIter-XL version of the code) or SD-XL based implementation feels a bit outdated. I guess you used it to implement based on IP-Adapter, but when we consider Flux.1.Kontext or Nano-banana-like reference-image based image editing models, I think these can also achieve good result in generating good result on such scenarios, so, I feel that current paper is outdated.
> I suggest adding the SOTA editing models (Flux.1.Kontext / Nano-Banana) into comparison. Since these model show good consistency, I think they can serve as powerful baselines. If your model performs somewhat close to these baselines, I think it will demonstrate the superiority of your method.
2) Question on (1) leads to questionable motivation. If we can achieve good result with Flux.1.Kontext or Nano-banana with good consistent result, why should we use this model to achieve good consistency? I see that the point of using Story-Iter framework is to achieve good quality of consistency with text alignment in iterative generation scenario, but if the SOTA models show good quality, I  cannot understand this paper's motivation.
> I want the motivation of the paper further strengthened.
3) Question on (1) and (2) leads to the question on the work's high computational cost: 10 external iterations for 100 frames requires 4.30 PFLOPs/25 minutes per iteration; total generation time is substantial despite claims of reducibility. Compared to those SOTA models, is it more efficient?
> How does performance scale with 3-5 iterations versus 10? What is the minimum number of iterations needed for acceptable quality-efficiency trade-off? Also I think adding comparison on inference FLOPs with those SOTA editing models is necessary.
4) Text-alignment degradation: Table 8 shows CLIP-T drops from 0.330 to 0.297 as iterations increase (from 1 to 15), indicating the method overfits to visual consistency at the expense of text fidelity
> What causes the text-alignment degradation (CLIP-T drop) with more iterations? Can this be addressed with alternative weighting strategies or architectures?
5) Limited diversity: Authors acknowledge "local frames still exhibit diversity limitations"; the iterative refinement may overly homogenize the visual sequence
> How does the method handle significant content changes across frames (scene transitions, new character introductions, location changes) with diverse scenarios? When you look at the result of Story-Iter, they look so-alike in terms of colors and backgrounds. We can say it is consistent, but it means less diverse, so I want some more explanations on how it demonstrates on complex/dynamic changing scenarios.
6) Incremental technical contribution: GRCA is essentially IP-Adapter's cross-attention adapted with global embeddings; the iterative paradigm, while effective, is a relatively straightforward extension. Moreover, doing the explicitly set iterations (10 for the paper) is somewhat naive.
> Even though the iterating over the result itself is somewhat interesting, I think there should be some way to stop at appropriate iteration if the desired quality is achieved.
7) Limited long-story evaluation: Custom benchmark has only 20 cases (10×50, 10×100 frames) generated by GPT-4o, which is insufficient for robust evaluation and may introduce generation biases
> It would be really good if we can evaluate on excessive dataset constructed since doing only on 20 cases can be a bit small.

**Questions:**

Written in the weakness.I know that (6) and (7) are difficult to answer during rebuttal period. Please focus on answering questions of (1) to (5). I will adjust my score after checking on the rebuttal.

---

> ### Author Response · Authors · 2025-11-20
> **Rebuttal by Authors (4/7)**
>
> We appreciate your constructive comments and recognition of our paper’s clarity and contribution. We address each concern below.
>
> ---
>
> >`Q1&Q2&Q3`: Comparison with reference-image based image editing models
>
> **A1&A2**: We added experiments in `Table 2`, `Section 4.3`, and `Appendix D`. While these models achieve strong identity consistency, they are unsuitable for long-story visualization:
>
> | Method            | Diffusion Steps | External Iterations | FLOPs       | VRAM | Time    | CLIP-T | aCCS  | aFID   |
> |-------------------|-----------------|----------------------|-------------|------|---------|--------|-------|--------|
> | StoryDiffusion    | 50              | 1                    | 22 PFLOPs   | 40GB | 31 min  | 0.315  | 0.768 | 102.44 |
> | Nano Banana | - | 1 | - | - | 33 min | 0.310 | 0.791 | 106.14 |
> |FLUX.1.Kontext | 50 | 1 | 371 PFLOPs | 68GB | 48 min | 0.290 | 0.783 | 78.03 |
> | Story-Iter        | 50              | 10                   | 43 PFLOPs   | 19GB | 250 min | 0.318  | 0.802 | 94.30  |
> | Story-Iter-Fast  | 4               | 10                   | 3 PFLOPs    | 19GB | 20 min  | 0.309  | 0.788 | 109.13 |
>
> - **(1&2.1)** **Overly strong identity constraints.** Editing models rigidly preserve reference appearance, causing more near-identical frames and degraded CLIP-T.
>
> - **(1&2.2)** **Cannot handle new entities.** They fail for characters/objects not present in the reference image.
>
> - **(1&2.3)** **Story-Iter supports open-ended narratives.** GRCA is training-free and content-adaptive, handling arbitrary scenes without fixed reference images.
>
> - **(1&2.4)** **Unfair comparison setting.** We note that comparing Story-Iter to commercial systems like Nano-Banana is not appropriate, as these models use far larger parameter scales, proprietary training data, and closed, unreproducible pipelines. Even under this inherent mismatch, Story-Iter—built entirely on transparent and fully reproducible open-source backbones—achieves comparable or even superior long-story consistency, showing that our gains come from the proposed methodology rather than model size.
>
> **A3**:
>
> - **(3.1)** **Efficiency.** Although Story-Iter uses fewer FLOPs and less VRAM than SOTA editing models, its external iterations lead to a higher inference time. To improve efficiency, we introduce a fast variant of Story-Iter based on the SDXL-LCM backbone (`Section 4.4` and `Appendix H.2`). It reduces diffusion steps from 50 to 4, substantially accelerating generation while maintaining story-level consistency.
>
> - **(3.2)** **Practical iteration.** Story-Iter achieves most of its gains within 3–5 iterations, reaching ~80% of the final consistency improvement (`Table 11`).
> Thus, 10 iterations are not required in practice; they were used in the paper to demonstrate the upper bound of consistency.
> A high-quality efficiency–performance trade-off can be obtained with only 3–5 iterations.
>
> | Iterations | CLIP-T | aCCS  | aFID   |
> |------------|--------|-------|--------|
> | 3          | 0.325  | 0.770 | 110.86 |
> | 5          | 0.319  | 0.783 | 100.81 |
> | 10         | 0.306  | 0.840 | 91.35  |
>
> ---
>
> >`Q4`: CLIP-T degradation
>
> **A4**: We thank the reviewer for pointing out the degradation in CLIP-T. We clarify that this behavior is mainly caused by the linear weighting schedule, not by a failure of the framework itself.
>
> - **(4.1)** The slight drop is due to the linear weighting, which gradually strengthens GRCA, thus weakening text attention.
>
> - **(4.2)** Similar to autoregressive decoding, global coherence increases while local sensitivity slightly decreases—but no collapse occurs (new objects/actions still generated; `Figure 24-Figure 33`).
>
> - **(4.3)** The trade-off is controllable: `Table 11` shows the early stopping yields trade-off.
>
> - **(4.4)** For future work (`Appendix G`), we plan to introduce Qwen3-VL–based content-aware scheduling, where Qwen3-VL evaluates the story semantics during refinement and adaptively decides early stopping and GRCA weight for next iterations.

---

> ### Author Response · Authors · 2025-11-24
> **Rebuttal by Authors (7/7)**
>
> We appreciate your constructive comments and recognition of our paper’s clarity and contribution. We address each concern below.
>
> ---
>
> >`Q5`: Handling scene transitions and diversity
>
> **A5**: We thank the reviewer for raising the concern regarding limited diversity.
>
> - **(5.1)** GRCA’s content-adaptive nature enables Story-Iter to handle new characters/objects and large scene transitions. Scene diversity is primarily dictated by text; GRCA propagates only relevant global context.
>
> - **(5.2)** At each iteration, GRCA selects reference frames based on semantic similarity (via pretrained IP-Adapter) and suppresses outdated/irrelevant content through soft forgetting by attention weight (see `Figure 3` and `Figure 12`).
>
> - **(5.3)** When large semantic changes occur, GRCA down-weights irrelevant scenes and focuses on the similar reference scene aligned with the new scene text due to high attention weight. So successful scene transitions appear in `Figure 24-Figure 33`. Background similarity occurs only when the story itself describes a stable setting.
>
> - **(5.4)** We also introduce Story-Iter-ControlNet in `Section 4.4` for explicit pose control, improving local diversity while retaining global consistency.
>
> ---
>
> >`Q6`: Incremental technical contribution for GRCA and early stopping
>
> **A6**: We agree with the reviewer that the main contribution of our paper lies in the proposed iterative paradigm rather than introducing a fundamentally new attention primitive.
>
> - **(6.1)** While GRCA is not a new attention primitive, but is also not a reuse of IP-Adapter. IP-Adapter supports single-frame subject consistency; GRCA supports multi-frame global story modeling.
>
> - **(6.2)** GRCA encodes all frames into global embeddings to build a story-level semantic representation that a single-image IP-Adapter cannot achieve.
>
> - **(6.3)** GRCA avoids autoregressive error accumulation by aggregating all frames, maintaining consistency for newly introduced characters.
>
> - **(6.4)** Ablations (`Table 4`, `Section A.6`, and `Figure 10`) show that removing GRCA severely reduces consistency, confirming its necessity.
>
> - **(6.5)** GRCA performs content-aware reference selection, suppressing irrelevant frames during scene transitions, as evidenced in `Figure 3`, `Figure 12`, and `Figure 24-Figure 33`.
>
> - **(6.6)** In future work (`Appendix G`), we plan to replace the fixed linear schedule with Qwen3-VL–based content-aware scheduling, where Qwen3-VL evaluates the story semantics during refinement and adaptively decides whether to stop iterating, and how to set the GRCA weight at the next step.
>
> ---
>
> >`Q7`: Limited long-story evaluation
>
> **A7**: We thank the reviewer for pointing out the limitation regarding the scale of long-story evaluation. We agree that a larger-scale evaluation would further strengthen the empirical validation. However, constructing meaningful long-form story benchmarks is substantially more challenging than typical vision–language datasets.
>
> - **(7.1)** Constructing long-story benchmarks is time-consuming, which requires coherent global plots, multi-character interactions, temporally aligned scene transitions, and precisely grounded descriptions. All long stories in our benchmark had to be manually designed, curated, and iteratively refined using GPT-based assistance, which makes large-scale construction extremely time-consuming.
>
> - **(7.2)** Our 20 curated cases cover diverse narrative structures and stress-test 50–100-frame consistency, which cover diverse narrative structures, multi-character interactions, complex scene transitions, and significant plot shifts.
>
> - **(7.3)** we have expanded the long-story evaluation in the **Supplementary Material**. Specifically, we augmented `story_list.py` with a large number of additional story cases, providing 1,500 newly added story prompts to further enrich the benchmark
>
> - **(7.4)** We plan to expand our benchmark and release a larger benchmark in future work (`Appendix G`).

---

> ### Author Response · Authors · 2025-11-25
> **Does Our Rebuttal Address Your Concerns?**
>
> To Reviewer LXWx,
>
> Thank you very much for your valuable comments and suggestions. We truly appreciate the time and effort you put into reviewing our paper.
>
> We have carefully addressed all the concerns you raised in our rebuttal.
>
> For example, **Q1&Q2**: Reference-image editing models cannot support long-story visualization because they enforce overly strong identity constraints, fail with new entities, and lack GRCA’s training-free, content-adaptive global story modeling. **Q3**: We address the efficiency concern by introducing the SDXL-LCM–based Story-Iter-Fast and show that practical usage requires only 3–5 iterations for strong performance. **Q4**: The CLIP-T drop is caused by the linear weighting schedule rather than the framework itself, and is mitigated through early stopping. Besides, we will consider a content-aware scheduling based on Qwen3-VL in the future. **Q5**: GRCA’s content-adaptive selection enables robust handling of new characters and scene transitions, and Story-Iter-ControlNet further improves local-frame diversity with explicit pose control. **Q6**: Although not a new attention primitive, GRCA is essential for global multi-frame modeling, avoiding autoregressive errors, and providing content-aware reference selection unattainable by a single-frame IP-Adapter. **Q7**: Long-story benchmark construction is inherently difficult and time-intensive, but our curated 20 cases already stress-test 50–100-frame consistency, and we plan to release a larger benchmark in future work.
>
> We would be grateful if you could take a moment to review our responses and let us know whether they sufficiently resolve your questions.
>
> We hope our replies have addressed your concerns, and we again sincerely appreciate your positive comments.
>
> We truly value your feedback and look forward to any further suggestions you may have.
>
> Best regards,
>
> The authors of Paper 13235

---

> ### Author Response · Authors · 2025-11-27
> **Follow-up on Rebuttal: Discussion Phase Closing Soon**
>
> Dear Reviewer, `LXWx`
>
> We sincerely appreciate your invaluable feedback, which has significantly contributed to the improvement of our work.
>
> Following your suggestions, we find reference-image editing models are unsuitable for long-story visualization because they impose overly strong identity constraints, fail on new entities `Table 2 Section 4.3 Appendix D`. Efficiency concerns are addressed by the SDXL-LCM–based Story-Iter-Fast `Section 4.4 Appendix H.2`, and practical usage typically requires only 3–5 iterations `Table 11`. The CLIP-T drop stems from the linear weighting schedule rather than the framework and can be mitigated via early stopping `Table 11`; future work will explore content-aware scheduling with Qwen3-VL `Appendix G`. GRCA enables reliable handling of new characters and scene transitions `Figure 3 Figure 12`, while Story-Iter-ControlNet adds explicit pose control to improve local diversity `Section 4.4 Figure 9`. Although not a new attention primitive, GRCA is essential for global multi-frame reasoning and effective reference selection beyond single-frame IP-Adapter behavior. Long-story benchmark construction is inherently challenging, but our curated 20 cases already stress-test 50–100-frame consistency, with plans to expand the benchmark in future work `Appendix G`.
>
> We hope that our revisions and clarifications have resolved your concerns. If you find our response satisfactory, we would be deeply grateful for a reconsideration of our score. Otherwise, if you have any additional questions, please do not hesitate to let us know. We would be more than willing to provide further clarification.
>
> We are truly grateful for your insightful comments, which have helped us improve the clarity and completeness of our work!
>
> Best regards,
>
> The Authors of Submission 13235

---

### Official Review · Reviewer_A9PP · 2025-11-01

**Soundness:** 3
**Presentation:** 4
**Contribution:** 3
**Rating:** 6
**Confidence:** 4

**Summary:**

This paper introduces Story-Iter, a novel, training-free iterative paradigm for long story visualization. The key problem addressed is maintaining semantic consistency and accurately rendering interactions in long image sequences generated from text prompts. Existing methods, primarily Auto-Regressive (AR) and Reference-Image (RI) paradigms, suffer from error accumulation or an inability to adapt to the full story context.

Story-Iter proposes an external iterative loop that refines the entire story sequence in each pass. In iteration `i`, the generation of each frame is conditioned on the text prompt *and* all generated frames from the previous iteration `i-1`. This is enabled by a plug-and-play Global Reference Cross-Attention (GRCA) module, which uses the global CLIP embeddings of all previous frames as context. The influence of this visual context is controlled by a linearly increasing weight $\\lambda_i$ across iterations, balancing text alignment and visual consistency. The authors demonstrate through extensive experiments on both regular-length (StorySalon) and a new long-story benchmark that Story-Iter achieves state-of-the-art performance in generating coherent and high-quality story visualizations, particularly for sequences up to 100 frames.

**Strengths:**

1.  **Novel and Effective Paradigm:** The core strength of the paper is the proposal of an iterative refinement paradigm for story visualization. This is a genuinely new approach in this domain that effectively addresses the key challenge of long-range consistency by allowing the model to gain a global view of the entire story and refine it over multiple passes. It elegantly sidesteps the error accumulation of AR models and the rigidity of RI models.

2.  **State-of-the-Art Performance:** The method achieves impressive empirical results, outperforming strong recent baselines like StoryDiffusion and StoryGen on benchmarks for both regular and long-story generation. The qualitative results, especially for long stories (up to 100 frames), are particularly compelling and clearly demonstrate superior character consistency and interaction modeling.

3.  **Excellent Presentation and Clarity:** The paper is extremely well-written, with clear explanations and high-quality figures that make the concepts easy to understand. The comprehensive appendix and extensive visual examples further bolster the quality of the submission.

4.  **Practicality and Accessibility:** By designing Story-Iter as a training-free, plug-and-play module that leverages pre-trained IP-Adapter weights, the authors have made their method easy to implement and use. This significantly lowers the barrier to entry for other researchers to verify, use, and extend this work.

**Weaknesses:**

1.  **Prohibitive Computational Cost:** The most significant weakness is the method's computational expense. An $L$-iteration process results in an $L$-fold increase in generation time compared to single-pass methods. The default of 10 iterations makes the method an order of magnitude slower than its competitors. This is a major practical limitation that is understated in the main paper and largely relegated to the appendix. While the authors suggest using acceleration techniques, no experiments are presented to show this is feasible without sacrificing quality. This trade-off between quality and compute needs to be a more central part of the discussion.

2.  **Limited Technical Novelty of GRCA:** The GRCA module is a direct reuse of the cross-attention mechanism and weights from IP-Adapter. While its application is novel, the module itself does not introduce a new architectural or algorithmic concept. The paper's contribution lies almost entirely in the iterative paradigm, not in the attention mechanism itself. This should be stated more explicitly to manage expectations about the technical depth of the proposed module.

3.  **Insufficient Justification of the Iterative Process:** The paper provides an intuitive but superficial explanation for why the iterative process works. The argument for convergence is based on a t-SNE visualization (Fig. 4), which is insufficient. A more formal discussion on the dynamics of this process would be beneficial. For example, what prevents the process from "over-fitting" to the initial generation and losing diversity, or collapsing to a mode where all images are nearly identical? The linear weighting $\\lambda_i$ is a heuristic to prevent this, but its behavior is not deeply analyzed.

4.  **Incomplete Ablation Studies:** The ablation study, while good, could be more comprehensive. The `w/o GRCA` ablation replaces global attention with a per-frame self-refinement, which is a very different task. A more informative ablation would be to compare GRCA against simpler aggregation strategies within the same iterative paradigm. For instance:
    *   What if only a sliding window of $k$ previous frames is used as reference?
    *   What if only a random subset of frames is used?
    *   What if the global embeddings `c_1...B` are simply mean-pooled into a single context vector?
    This would help to truly justify the need for attending to *all* frames in the sequence.

**Questions:**

1.  **Computational Cost:** Could you please add a table to the main paper that explicitly compares the end-to-end inference time (or total GFLOPs) and peak VRAM usage of Story-Iter (10 iterations) against StoryDiffusion and StoryGen for generating a 100-frame story? This would provide a clearer picture of the quality-compute trade-off.

2.  **Dynamics of Iteration:** The linearly increasing weight $\\lambda_i$ is crucial for balancing text-alignment and consistency. Is there a risk that in later iterations, the strong visual conditioning from `GRCA` overpowers the text prompt $T_k$, leading to a failure to generate new objects or actions described in the text? The slight drop in CLIP-T in Table 8 suggests this might be the case. How does the model handle the introduction of entirely new characters or settings late in the story across iterations?

3.  **Metric Definition:** For reproducibility and clarity, could you please add formal definitions of `aCCS` and `aFID` to the appendix? Specifically, how are character bounding boxes obtained for `aCCS` calculation, and is it robust to detection failures? For `aFID`, is the distance computed between all pairs of images, consecutive images, or against a reference set?

4.  **Sensitivity to Hyperparameters:** The linear schedule for $\\lambda_i$ is defined by $\\lambda_1$ and $\\lambda_L$. How sensitive is the final result to these start/end points? Does the optimal schedule depend on the story length $B$ or the diversity of the content within the story? For example, would a story with many scene changes require a different schedule than one with a fixed background?

5.  **Necessity of Full Context:** The GRCA module attends to all $B$ frames from the previous iteration. Have you investigated whether this is necessary? Could a similar level of performance be achieved with a more efficient context, such as a fixed-size window of $k$ surrounding frames, or a stochastically sampled subset of frames? This could be a path to mitigating the high computational cost.

6.  **Potential for Training:** The training-free aspect is a great feature for accessibility. However, have you considered the possibility that fine-tuning the IP-Adapter weights (or a dedicated GRCA module) on the task of iterative story refinement could lead to even better performance or, more importantly, reduce the number of required iterations, thus improving efficiency?

---

> ### Author Response · Authors · 2025-11-24
> **Rebuttal by Authors (7/8)**
>
> Thank you for your thoughtful and positive feedback. We address your concerns point by point.
>
> ---
>
> > `Q1`: Prohibitive Computational Cost.
>
> **A1**: To improve efficiency, we introduce a fast variant of Story-Iter based on the SDXL-LCM backbone (`Section 4.4`; `Appendix H.2`). It reduces diffusion steps from 50 to 4, substantially accelerating generation while maintaining story-level consistency.
>
> | Method            | Diffusion Steps | External Iterations | FLOPs       | VRAM | Time    | CLIP-T | aCCS  | aFID   |
> |-------------------|-----------------|----------------------|-------------|------|---------|--------|-------|--------|
> | StoryDiffusion    | 50              | 1                    | 22 PFLOPs   | 40GB | 31 min  | 0.315  | 0.768 | 102.44 |
> | Story-Iter        | 50              | 10                   | 43 PFLOPs   | 19GB | 250 min | 0.318  | 0.802 | 94.30  |
> | Story-Iter-Fast | 4               | 10                   | 3 PFLOPs    | 19GB | 20 min  | 0.309  | 0.788 | 109.13 |
>
> ---
>
> > `Q2`: Limited Technical Novelty of GRCA.
>
> **A2**: We agree that the main contribution is the iterative paradigm, yet GRCA is not a trivial reuse of IP-Adapter.
>
> - **(2.1)** IP-Adapter’s cross-attention targets single-frame consistency, whereas GRCA is designed for multi-frame story visualization with very different requirements.
>
> - **(2.2)** GRCA is the first to encode all frames into global embeddings and use them jointly, forming global story comprehension rather than single-image matching.
>
> - **(2.3)** GRCA aggregates all reference frames (not only past ones), preventing AR-style error accumulation and maintaining new-character consistency.
>
> - **(2.4)** Ablation results (`Table 4`, `Section A.6`, and `Figure 10`) show that replacing GRCA with a single-frame reference drastically degrades consistency.
>
> - **(2.5)** GRCA performs content-aware reference selection, strengthening semantically aligned frames and suppressing irrelevant ones (see `Figure 3` and `Figure 12`). This enables coherent transitions during scene/plot changes in `Figure 24-Figure 33`.
>
> ---
>
> > `Q3`: Insufficient Justification of the Iterative Process.
>
> **A3**: We thank the reviewer for raising this important question.
>
> - **(3.1)** GRCA is content-adaptive and recomputes attention at each iteration, avoiding over-fitting to the initial generation (see `Figure 3` and `Figure 12`).
>
> - **(3.2)** Story-Iter does not collapse because each frame still conditions on its own text prompt; GRCA only supplies identity/style coherence.
>
> - **(3.3)** Linear λᵢ is not the stabilizer; stability comes from: (1) GRCA’s adaptive attention; (2) independent text conditioning per frame; (3) residual fusion. `Figure 8` shows: λ too small → no improvement; λ too large → collapse. Stability arises from GRCA behavior.
>
> - **(3.4)** `Table 11` shows convergence after ~8–10 iterations: consistency improves, then plateaus; text alignment drops slightly but does not collapse.
>
> ---
>
> > `Q4`: Incomplete Ablation Studies for GRCA.
>
> **A4**: Following the reviewer’s suggestion, we conducted ablations in `Appendix A.6`:
>
> - **(4.1)** Sliding window captures only short-range consistency; fails on long-range reappearance.
>
> - **(4.2)** Random subset introduces unstable cues; error accumulates.
>
> - **(4.3)** Mean pooling loss semantic information.
>
> | Setting                     | CLIP-T | aCCS  | aFID   |
> |----------------------------|--------|-------|--------|
> | Sliding Window             | 0.313  | 0.774 | 101.08 |
> | Random Reference Set       | 0.306  | 0.750 | 119.52 |
> | Mean-pooled Global Embedding | 0.311  | 0.741 | 108.19 |
> | **Story-Iter**             | **0.318**  | **0.802** | **94.30** |
>
> ---
>
> > `Q5`: Dynamics of Iteration.
>
> **A5**: Increasing λᵢ does not overpower text or block new concepts, because:
>
> - **(5.1)** Text prompt is re-injected at every iteration.
>
> - **(5.2)** All reference frames originate from text-conditioned synthesis.
>
> - **(5.3)** Newly introduced objects/characters are reintroduced in later iterations’ GRCA.
>
> - **(5.4)** GRCA is selective: it amplifies relevant frames and suppresses outdated ones; λᵢ strengthens global coherence but does not erase new semantics.
>
> `Figure 24-Figure 33` shows successful late-appearing characters/scenes.
>
> ---
>
> > `Q6`: Metric Definition.
>
> **A6**: Definitions of aCCS and aFID are included in `Appendix B`.
>
> ---
>
> > `Q7`: Sensitivity to Hyperparameters.
>
> **A7**: $ \lambda_1 $ and $ \lambda_L $ affect the rate of introducing consistency, but not sensitivity to story length or content diversity. GRCA plays the dominant role:
>
> - **(7.1)** It dynamically focuses on semantically aligned reference frames (see `Figure 3` and `Figure 12`).
>
> - **(7.2)** Irrelevant history is suppressed via soft forgetting by GRCA attention weight.
>
> - **(7.3)** Linear schedule mainly controls consistency speed; simple tuning or early stopping yields good trade-offs.
>
> We plan to propose a Qwen3-VL–based content-aware schedule (`Appendix G`).

---

> ### Author Response · Authors · 2025-11-24
> **Rebuttal by Authors (8/8)**
>
> Thank you for your thoughtful and positive feedback. We address your concerns point by point.
>
> ---
>
> > `Q8`: Potential for Training.
>
> **A8**: We appreciate the reviewer’s insightful suggestion. However:
>
> - **(8.1)** No large, copyright-free long-story datasets exist for supervised fine-tuning.
>
> - **(8.2)** Fine-tuning risks harming Story-Iter’s open-domain generalization.
>
> - **(8.3)** We already provide Story-Iter-Fast in `Section 4.4` and `Appendix H.2` to solve the efficiency issue.

---

> ### Author Response · Authors · 2025-11-25
> **Does Our Rebuttal Address Your Concerns?**
>
> To Reviewer A9PP,
>
> Thank you very much for your valuable comments and suggestions. We truly appreciate the time and effort you put into reviewing our paper.
>
> We have carefully addressed all the concerns you raised in our rebuttal.
>
> For example, **Q1**: We improve efficiency by introducing the SDXL-LCM–based Story-Iter-Fast variant, which reduces diffusion steps from 50 to 4 and substantially accelerates generation while preserving story consistency. **Q2**: GRCA provides nontrivial technical value by enabling global multi-frame embedding, content-aware reference selection, and long-range identity/style coherence far beyond single-frame IP-Adapter behavior. **Q3**: The iterative process is justified because GRCA adaptively recalibrates reference attention each round, combined with per-frame text conditioning and residual fusion that stabilize refinement without collapse. **Q4**: Our extended ablations demonstrate that alternatives like sliding windows, random subsets, or mean pooling fail to maintain long-range semantic consistency, confirming GRCA’s necessity. **Q5**: Increasing λᵢ does not suppress new semantics because text prompts are injected at every iteration and GRCA selectively amplifies relevant frames while naturally forgetting outdated ones. **Q6**: The definitions of aCCS and aFID are fully provided in Appendix B. **Q7**: Linear weighting strategy mainly controls the speed of injecting consistency, while GRCA’s content-adaptive attention remains the dominant stabilizing factor across different story lengths and complexities. **Q8**: Training is not pursued due to the absence of large copyright-free long-story datasets and the risk of harming open-domain generalization, and efficiency concerns are already addressed by Story-Iter-Fast.
>
> We would be grateful if you could take a moment to review our responses and let us know whether they sufficiently resolve your questions.
>
> We hope our replies have addressed your concerns, and we again sincerely appreciate your positive comments.
>
> We truly value your feedback and look forward to any further suggestions you may have.
>
> Best regards,
>
> The authors of Paper 13235

---

> ### Author Response · Authors · 2025-11-27
> **Follow-up on Rebuttal: Discussion Phase Closing Soon**
>
> Dear Reviewer, `A9PP`
>
> We sincerely appreciate your invaluable feedback, which has significantly contributed to the improvement of our work.
>
> Following your suggestions, we improved efficiency by introducing the SDXL-LCM–based Story-Iter-Fast variant, which reduces diffusion steps from 50 to 4 and substantially accelerates generation while preserving story consistency `Section 4.4 Appendix H.2`. GRCA provides nontrivial technical value by enabling global multi-frame embedding, content-aware reference selection, and long-range identity/style coherence far beyond the single-frame IP-Adapter. The iterative process is justified because GRCA adaptively recalibrates reference attention each round, combined with per-frame text conditioning and residual fusion that stabilize refinement without collapse. Our extended ablations demonstrate that alternatives like sliding windows, random subsets, or mean pooling fail to maintain long-range semantic consistency, confirming GRCA’s necessity `Appendix A.6`. Increasing λᵢ does not suppress new semantics because text prompts are injected at every iteration and GRCA selectively amplifies relevant frames while naturally forgetting irrelevant ones `Figure 3 Figure 12`. The definitions of aCCS and aFID are fully provided in `Appendix B`. Linear weighting strategy mainly controls the speed of injecting consistency, while GRCA’s content-adaptive attention remains the dominant stabilizing factor across different story lengths and complexities. Training is not pursued due to the absence of large copyright-free long-story datasets and the risk of harming open-domain generalization, and efficiency concerns are already addressed by Story-Iter-Fast `Section 4.4 Appendix H.2`.
>
> We hope that our revisions and clarifications have resolved your concerns. If you find our response satisfactory, we would be deeply grateful for a reconsideration of our score. Otherwise, if you have any additional questions, please do not hesitate to let us know. We would be more than willing to provide further clarification.
>
> We are truly grateful for your insightful comments, which have helped us improve the clarity and completeness of our work!
>
> Best regards,
>
> The Authors of Submission 13235

---

### Official Review · Reviewer_TFke · 2025-11-01

**Soundness:** 3
**Presentation:** 3
**Contribution:** 3
**Rating:** 6
**Confidence:** 4

**Summary:**

This paper proposes Story-Iter, a training-free iterative paradigm for long story visualization that addresses the limitations of existing auto-regressive and reference-image paradigms by using full-length frames from the previous external iteration as references and integrating a Global Reference Cross-Attention (GRCA) module to model global semantic consistency.

**Strengths:**

1. **Novel and Targeted Iterative Paradigm**: The proposed external iterative framework directly addresses the core limitations of existing AR and RI paradigms in long story visualization. By using full-length frames from the previous iteration as references (instead of fixed or limited frames), it effectively mitigates error accumulation and global consistency loss, a long-standing challenge in the field .

2. **Efficient and Lightweight GRCA Module**: The Global Reference Cross-Attention (GRCA) module, based on CLIP global embeddings, enables global semantic modeling while significantly reducing computational costs.

3. **Rigorous Experimental Validation**: Experiments on both regular-length (StorySalon) and long-story (100-frame) benchmarks show consistent SOTA performance: it outperforms StoryGen and surpasses StoryDiffusion for long stories. Human evaluations further confirm its superiority in character interaction and content consistency .

4. **Practical Training-Free Design**: Story-Iter requires no retraining, leveraging pre-trained Stable Diffusion and CLIP weights. Its plug-and-play GRCA and linear weighting strategy ensure easy integration into existing pipelines, with minimal hyperparameter tuning .

5. **Strong Generalization to Multi-Style Generation**: Beyond standard realistic style, it successfully generates long stories in comic, film, and monochrome styles, with CLIP-T scores remaining above 0.30. This demonstrates robust adaptability to diverse visual requirements .

**Weaknesses:**

1. **Computational Efficiency for Extremely Long Stories**: While more efficient than baselines, generating a 100-frame 1024×1024 story still incurs 4.30 PFLOPs per iteration. If the paper were to include experiments based on a distilled single-step diffusion model to demonstrate the universality of its method, it would more fully prove the superiority of its method.

2. **Tradeoff Between Consistency and Text Alignment**: Longer iterations (≥10) slightly weaken text-image alignment (CLIP-T drops from 0.330 to 0.297), and the current linear weighting strategy lacks content-aware adaptability for stories with complex plot shifts .

3. **Limited Diversity in Local Frames**: Despite global consistency, certain consecutive frames exhibit limited diversity (e.g., similar character poses). The framework lacks explicit controls (e.g., pose/layout constraints) to enhance frame-wise variation .

**Questions:**

Please refer to the detailed points I raised in the "Weakness" section and respond to each numbered item in your rebuttal with clarifications.

---

> ### Author Response · Authors · 2025-11-20
> **Rebuttal by Authors**
>
> We sincerely appreciate your constructive comments and your recognition of our paper’s clarity and contribution. We will explain your concerns point by point.
>
> ---
>
> > `Q1`: Computational Efficiency for Extremely Long Stories.
>
> **A1**: We sincerely appreciate your question. To address the limitation regarding the computational efficiency of Story-Iter for long-story visualization, we introduce a fast variant of Story-Iter built upon the SDXL-LCM backbone in `Section 4.4` and `Appendix H.2`. This version reduces the diffusion sampling steps from 50 to only 4, drastically improving the iteration efficiency while maintaining comparable story-level consistency.
>
> | Method            | Diffusion Steps | External Iterations | FLOPs       | VRAM | Time    | CLIP-T | aCCS  | aFID   |
> |-------------------|-----------------|----------------------|-------------|------|---------|--------|-------|--------|
> | StoryDiffusion    | 50              | 1                    | 22 PFLOPs   | 40GB | 31 min  | 0.315  | 0.768 | 102.44 |
> | Story-Iter        | 50              | 10                   | 43 PFLOPs   | 19GB | 250 min | 0.318  | 0.802 | 94.30  |
> | Story-Iter-Fast | 4               | 10                   | 3 PFLOPs    | 19GB | 20 min  | 0.309  | 0.788 | 109.13 |
>
> These results demonstrate that the proposed GRCA and the iterative framework are fully compatible with distilled and accelerated diffusion models, and that Story-Iter can maintain strong global story coherence even under extremely efficient sampling regimes.
>
> ---
>
> > `Q2`: Tradeoff Between Consistency and Text Alignment.
>
> **A2**: We appreciate the reviewer’s comment regarding the trade-off between consistency and text–image alignment in external iteration.
>
> - **(2.1)** We would like to clarify that complex plot or scene transitions in Story-Iter are not totally governed by the linear weighting schedule, but rather are mainly handled by the content-adaptive mechanism inside GRCA. Specifically, GRCA inherently performs content-aware reference selection: It dynamically adjusts attention weight over reference frames based on pretrained IP-Adapter features. It prioritizes reference frames that are semantically aligned with the current target frame. Irrelevant or outdated frames are naturally suppressed through the soft forgetting behavior of the global embedding. Thus, when large semantic changes occur (e.g., sudden scene shifts or plot transitions), Story-Iter does not propagate irrelevant history, but instead focuses on the subset of reference frames that remain meaningful for the new scene. This enables Story-Iter to maintain coherent transitions across complex narratives, as also evidenced in `Figure 3`, `Figure 12`, and qualitative examples in `Figure 24-Figure 33`.
>
> - **(2.2)** Regarding the observed trade-off between long-iteration consistency and text–image alignment, this phenomenon is analogous to autoregressive models: longer inference allows stronger global coherence but may attenuate local fidelity to the immediate text prompt. Importantly, this does not indicate inefficiency in the iterative refinement process but reflects an inherent balancing between global story structure and local prompt specificity. This trade-off is also effectively mitigable. As shown in `Table 11`, appropriate hyperparameter tuning—adjusting iteration count based on story length/content and applying early stopping—already provides a better balance. We add a promising solution in `Appendix G`, which we plan to introduce Qwen3-VL–based content-aware scheduling, where Qwen3-VL evaluates story semantics during refinement and adaptively decides whether an iteration should stop early and set the next iteration’s GRCA/text-conditioned weighting. This content-adaptive controller will further optimize consistency–alignment trade-offs without sacrificing Story-Iter’s strong narrative coherence.
>
> ---
>
> > `Q3`: The framework lacks explicit controls (e.g., pose/layout constraints) to enhance frame-wise variation.
>
> **A3**: Thank you for the constructive suggestion!
>
> - **(3.1)** To directly address this concern, we introduce a new variant Story-Iter-ControlNet in `Section 4.4`, which integrates a pose-conditioned ControlNet into the Story-Iter. This variant provides explicit control over character pose and spatial layout, while fully preserving the global story consistency and text–image alignment achieved by Story-Iter.
>
> - **(3.2)** As shown in `Figure 9`, using pose maps (e.g., OpenPose skeletons) as control signals, Story-Iter-ControlNet offers:
>   - greater intra-scene diversity across consecutive frames (e.g., distinct actions and poses);
>   - fine-grained and controllable manipulation of character posture;
>   - stable global consistency and text alignment;
>   - no degradation in long-story coherence;
>
> Therefore, Story-Iter can naturally incorporate explicit structural guidance when needed, further improving local-frame diversity without compromising global fidelity.

---

> ### Author Response · Authors · 2025-11-25
> **Does Our Rebuttal Address Your Concerns?**
>
> To Reviewer TFke,
>
> Thank you very much for your valuable comments and suggestions. We truly appreciate the time and effort you put into reviewing our paper.
>
> We have carefully addressed all the concerns you raised in our rebuttal.
>
> For example, **Q1**: we address efficiency concerns by introducing the SDXL-LCM–based Story-Iter-Fast variant, which reduces diffusion steps from 50 to 4 and greatly accelerates external iterations while maintaining strong story-level consistency. **Q2**: GRCA’s content-adaptive reference selection naturally preserves semantic transitions, and our planned content-aware scheduling further balances global coherence with local text alignment. **Q3**: We incorporate explicit structural control through Story-Iter-ControlNet, which adds pose-conditioned guidance to enhance frame-wise diversity without sacrificing global story coherence.
>
> We would be grateful if you could take a moment to review our responses and let us know whether they sufficiently resolve your questions.
>
> We hope our replies have addressed your concerns, and we again sincerely appreciate your positive comments.
>
> We truly value your feedback and look forward to any further suggestions you may have.
>
> Best regards,
>
> The authors of Paper 13235

---

> ### Author Response · Authors · 2025-11-27
> **Follow-up on Rebuttal: Discussion Phase Closing Soon**
>
> Dear Reviewer, `TFke`
>
> We sincerely appreciate your invaluable feedback, which has significantly contributed to the improvement of our work.
>
> Following your suggestions, we address efficiency by introducing a fast SDXL-LCM variant that reduces diffusion steps from 50 to 4 while preserving story-level consistency, `Section 4.4 Appendix H.2`. GRCA ensures smooth semantic transitions through content-adaptive reference selection `Figure 3 Figure 12 Figure 24-Figure 33`, and future content-aware scheduling will be designed to further balance global coherence and local alignment `Appendix G`. Additionally, Story-Iter-ControlNet provides explicit structural control via pose-conditioned guidance, improving frame-level diversity without compromising narrative coherence `Section 4.4 Figure 9`.
>
> We hope that our revisions and clarifications have resolved your concerns. If you find our response satisfactory, we would be deeply grateful for a reconsideration of our score. Otherwise, if you have any additional questions, please do not hesitate to let us know. We would be more than willing to provide further clarification.
>
> We are truly grateful for your insightful comments, which have helped us improve the clarity and completeness of our work!
>
> Best regards,
>
> The Authors of Submission 13235

---

### Author Response · Authors · 2025-11-20
**General Response**

**Dear Reviewers, ACs, and SACs,**

We deeply appreciate the insightful and valuable comments provided by all reviewers.

---

We are grateful for all reviewers' recognition of this work as a **genuinely new approach in story visualization** that effectively addresses the key challenge of long-range consistency by allowing the model to gain a global view of the entire story and refine it over multiple passes. It elegantly sidesteps the error accumulation of AR models and the rigidity of RI models.

Overall, we are encouraged by the reviewers' positive feedback, which highlights:

- **Novel iterative refinement paradigm** that effectively addresses long-range consistency and avoids the limitations of AR and reference-image methods (Reviewers `TFke`, `A9PP`, `LXWx`).

- **Scalable and efficient GRCA module** enabling 100+ frame stories with manageable memory (Reviewers `TFke`, `LXWx`).

- **Strong empirical performance** with comprehensive experiments, ablations, and qualitative results demonstrating superior fine-grained interactions and consistency (Reviewers `TFke`, `A9PP`, `LXWx`).

- **Training-free, plug-and-play design** that reuses pretrained IP-Adapter weights and is easy to adopt and extend (Reviewers `TFke`, `A9PP`, `LXWx`).

To address the reviewers' concerns, we have conducted several additional experiments or designs, including:

- **Story-Iter-Fast Variant** to solve computational efficiency for extremely long stories with limited degradation (Reviewers `TFke`, `A9PP`, `LXWx`).

- **Story-Iter-ControlNet** to enhance frame-wise variation (Reviewers `TFke`, `A9PP`, `LXWx`).

- **More GRCA ablation studies** to highlight GRCA contribution in global story consistency (Reviewers `TFke`, `A9PP`, `LXWx`).

- **Comparison with reference-image based image editing models** to support Story-Iter motivation in long story visualization (Reviewer `TFke`).

---

**Summary of revisions:**

- Two variants of Story-Iter in `Section 4.4` and `Appendix H.2`, corresponding codes are provided in the **Supplementary Material** `run_controlnet.py` and `run_fast.py`

- Added comparison results with Flux.1.Kontext / Nano-Banana  in updated `Table 2`, `Section 4.3`, and `Appendix D`

- Added more GRCA ablation studies in `Appendix A.6`

- Updated the more detailed evaluation metric definition in  `Appendix B`

- Updated future work for balance consistency and text–image alignment, and expanded long story datasets in `Appendix G`

- Updated the 3rd Iteration results in  `Table 11` to show performance scale with 3-5 iterations versus 10

- Updated extra long story cases in `story_list.py` of Supplementary Material.

All revisions in the paper are highlighted in **blue**. We sincerely appreciate the reviewers' constructive suggestions and remain committed to continually improving our work.

---

We address each reviewer's comments point by point below. We welcome further discussion and look forward to continued engagement. Thank you!

---

### Meta-Review · Area_Chair_9rBx · 2025-12-11

**Summary:**

After reading the paper and rebuttal, I am inclined to accept this paper.  The strength slightly outweighs the weakness. However, I am ok if this paper is rejected.

Here is a summary of the concerns.

1. Computational Efficiency for Extremely Long Stories (Reviewer TFke, A9PP).
2. Limited Diversity in Local Frames (Reviewer TFke,LXWx).
3.  Incomplete Ablation Studies for GRCA (A9PP).
4. Base model (IP-Adapter) outdated and questionable motivation. (Reviewer LXWx)
5. Limited Technical Novelty (A9PP, LXWx)
6. Limited long-story evaluation: Custom benchmark has only 20 cases (10×50, 10×100 frames) generated by GPT-4o, which is insufficient for robust evaluation and may introduce generation biases (LXWx)

**Reviewer Concerns:**

1. TFke (mostly solved).
2. A9PP (partially solved): Incomplete Ablation Studies and Computational Efficiency solved. Limited technical novelty remains.
3. LXWx (partially solved): however, I agree that the baseline model is outdated and the proposed method is quite limited to old SD models.

**Reviewer Scores:**

1. TFke would keep the initial score.
2. A9PP would keep the initial score.
3. LXW would keep the initial score.

Therefore, this is a very borderline paper.

---

### Decision · Program_Chairs · 2026-01-26

Accept (Poster)